# Chromatin swelling drives neutrophil extracellular trap release

Elsa Neubert[1,2], Daniel Meyer[2,3], Francesco Rocca[3,4], Gökhan Günay[1,2], Anja Kwaczala-Tessmann[1], Julia Grandke[1], Susanne Senger-Sander[1], Claudia Geisler[3,4], Alexander Egner[3,4], Michael P. Schön [1,5], Luise Erpenbeck [1] & Sebastian Kruss [2,3]

Neutrophilic granulocytes are able to release their own DNA as neutrophil extracellular traps (NETs) to capture and eliminate pathogens. DNA expulsion (NETosis) has also been documented for other cells and organisms, thus highlighting the evolutionary conservation of this process. Moreover, dysregulated NETosis has been implicated in many diseases, including cancer and inflammatory disorders. During NETosis, neutrophils undergo dynamic and dramatic alterations of their cellular as well as sub-cellular morphology whose biophysical basis is poorly understood. Here we investigate NETosis in real-time on the single-cell level using fluorescence and atomic force microscopy. Our results show that NETosis is highly organized into three distinct phases with a clear point of no return defined by chromatin status. Entropic chromatin swelling is the major physical driving force that causes cell morphology changes and the rupture of both nuclear envelope and plasma membrane. Through its material properties, chromatin thus directly orchestrates this complex biological process.

[1] Department of Dermatology, Venereology and Allergology, University Medical Center, Goettingen University, Göttingen 37075, Germany. [2] Institute of Physical Chemistry, Göttingen University, Göttingen 37077, Germany. [3] Center for Nanoscale Microscopy and Molecular Physiology of the Brain (CNMPB), Göttingen 37073, Germany. [4] Optical Nanoscopy, Laser-Laboratorium Göttingen e.V., Göttingen 37077, Germany. [5] Lower Saxony Institute of Occupational Dermatology, University Medical Center Göttingen and University of Osnabrück, Göttingen 37075, Germany. These authors contributed equally: Elsa Neubert, Daniel Meyer. Correspondence and requests for materials should be addressed to L.E. (email: luise.erpenbeck@med.uni-goettingen) or to S.K. (email: skruss@uni-goettingen.de)

Neutrophilic granulocytes are the most abundant immune cells in humans and essential to defeat invading pathogens[1]. Their mechanisms to target invading microbes include well-known processes such as phagocytosis and generation of reactive oxygen species (ROS). A third defense pathway is the release of neutrophil extracellular traps (NETs)[2]. The formation of NETs (NETosis) can be triggered by organisms such as bacteria or different chemicals and was originally described as an additional form of cell death apart from apoptosis and necrosis[3–5]. NETosis has been reported not only for neutrophils but also other immune cells[6,7], amoebas[8] and plant cells[9] indicating an evolutionary conserved process[3].

During NETosis, cells can release three-dimensional meshworks (NETs) consisting of chromatin[2], antimicrobial components including myeloperoxidase (MPO)[5], neutrophil elastase (NE)[10], and LL37 of the cathelicidin family[11]. These fibrous networks were initially described as a mechanism to catch and eliminate bacteria, fungi, as well as viral particles[2]. However, it is becoming increasingly clear that the role of NETs in the immune system is far more complex than originally estimated. On the one hand, accumulating data suggests that the immediate role of NETs in immunoprotection against pathogens may be smaller than originally anticipated, as mice that cannot form NETs do not suffer from severe immunosuppression[12,13]. On the other hand, dysregulated or excessive NETosis appears to be implicated in an ever growing number of diseases, including cancer[14], thrombosis and vascular diseases[15–17], preeclampsia[18], chronic inflammatory diseases[19], and ischemia-reperfusion injury after myocardial infarction[16].

Various stimuli such as bacteria, fungi, viruses, platelets, as well as small compounds including lipopolysaccharides (LPS), calcium ionophores (CaI), or phorbol-myristate acetate (PMA) induce NETosis and release of NETs[20]. In many settings, NETosis appears to rely on the adhesion of neutrophils, in particular on the engagement of neutrophilic integrin receptors such as Mac-1[21–23], in others, adhesion via Mac-1 seems to be dispensable[24–26]. It has also been described that hemodynamic forces can trigger shear-induced NETosis[27].

While these triggers—biochemical or mechanical—engage diverse pathways, they all converge to a uniform outcome, namely histone modification, chromatin decondensation and NET release[28]. Cells dramatically rearrange their contents (cytoskeleton, organelles, membranes, nucleus) during NETosis; in most scenarios, they eventually die[4]. Chromatin decondensation has been described qualitatively since the discovery of NETs[4,29,30] and NET formation has been evaluated both in high-throughput approaches, as well as on the single-cell level[29–31]. Yet, the mechanistic basis of these fundamental changes, as well as the underlying dynamic forces remain poorly characterized. Here, we investigate NETosis from a biophysical perspective, particularly looking at the forces and dynamics driving this process, and provide functional links between chromatin dynamics and biochemical behavior. We show that NETosis is organized into well-defined phases orchestrated by entropic swelling of chromatin, which finally ruptures the membrane.

## Results

**NETosis is organized into distinct phases**. To better understand how the cell's interior is rearranged and how NETs are released we studied human neutrophils in real-time. First, we imaged chromatin and cell membranes of human neutrophils stimulated by 100 nM PMA (Fig. 1a, b, Supplementary Movies 1, 2). NETosis was confirmed by co-localization of chromatin and MPO within the expelled NETs (Fig. 1f).

The chromatin-filled area inside the cells followed a characteristic time course (Fig. 1a, b) that consistently allowed the assignment of three distinct phases. As can be seen later this phase classification allows us to distinguish active biological processes from materials driven processes and to identify a point of no return. Cells were stimulated ($t = 0$) and during the first phase P1 ($0 < t < t_1 =$ start of chromatin expansion) the lobular structure of the nucleus was still intact (34 min in Fig. 1a) and the corresponding chromatin area stayed constant. In the second phase P2 ($t_1 < t < t_3 =$ NET release) chromatin expanded within a few minutes until it reached the cell membrane as a barrier ($t_2 =$ maximal chromatin expansion in Fig. 1a). Simultaneously, the cell rounded up ($t_2$ to $t_3$ in Fig. 1a and Supplementary Fig. 1a, b). In the third and final phase (P3, $t > t_3$) the cell membrane is ruptured ($t_3$) and the NET released into the extracellular space. Additionally, released NETs were also visualized by stimulated emission depletion (STED) nanoscopy to reveal the architecture of hydrated NETs below the resolution limit of normal fluorescence microscopy (Supplementary Fig. 2). The observed architecture was in good agreement with previous electron microscopy images of NETs[2].

The above-mentioned phase classification was applied to multiple cells ($n = 139$ cells) from five different donors (Fig. 1c, Supplementary Fig. 3a-c, Supplementary Movies 3–6). Although the onset time points for the different phases of all individual cells followed a broad distribution (Fig. 1c), average onset values for all five donors were remarkably reproducible, indicating low interindividual variability for the three distinct phases under standardized conditions (Supplementary Fig. 3c). Decondensation of chromatin started at $t_1 = 56 \pm 4$ min (standard error of the mean, SEM) and reached a maximum at $t_2 = 82$ min ($\pm 3$ min). After $t_3 = 116$ min ($\pm 4$ min) the cytoplasmic membrane ruptured and the NETs were released. In summary, P1, as well as P2, require around 60 min in our experimental setup.

The three distinct phases of chromatin decondensation were not only elicited by PMA but also by LPS (Fig. 1d) or calcium ionophores (Fig. 1e), albeit with different onset times, particularly with respect to P1 (Supplementary Fig. 3d, Supplementary Movie 7), which is most likely an expression of the distinct signaling pathways engaged by different stimuli[28].

To understand biophysical events during NETosis in detail, we analyzed chromatin shape with special regard to the most relevant internal and external boundaries (i.e., nuclear envelope and cell membrane). In P1, cells first adhered to the surface, flattened and showed filopodia activity (Supplementary Fig. 4a, b, Supplementary Movies 1, 2, 8). During P2, the cell retracted its cell body as shown by CLSM (Fig. 1a, Supplementary Fig. 4a, b) and time-resolved reflection interference contrast microscopy (RICM) (Supplementary Fig. 4c, d and Supplementary Movies 9–11), the cells rounded up and their height increased as demonstrated by both three-dimensional CLSM stacks (side view, Fig. 2a) and atomic force microscopy (AFM) (Fig. 2b).

Interestingly, NETotic cells reached approximately the same height as non-stimulated round cells (Fig. 2b), suggesting that chromatin expansion/decondensation is a swelling process and the swelling pressure caused the energetically most favorable spherical shape.

To address the question of how chromatin exits the nucleus, we labeled lamin B1, a constitutive component of the nuclear envelope that surrounded all individual lobuli of the nucleus (Fig. 2c, Supplementary Fig. 5). Over time, those lobuli merged and became less distinctive, indicating profound changes in the separation between chromatin. Around the onset of P2 ($t_1$; i.e., the start of chromatin expansion), the lamin B1 layer tore on at least one site of the nucleus. Often, several rupture events of this layer were discernable (arrows Fig. 2c, see also comparison

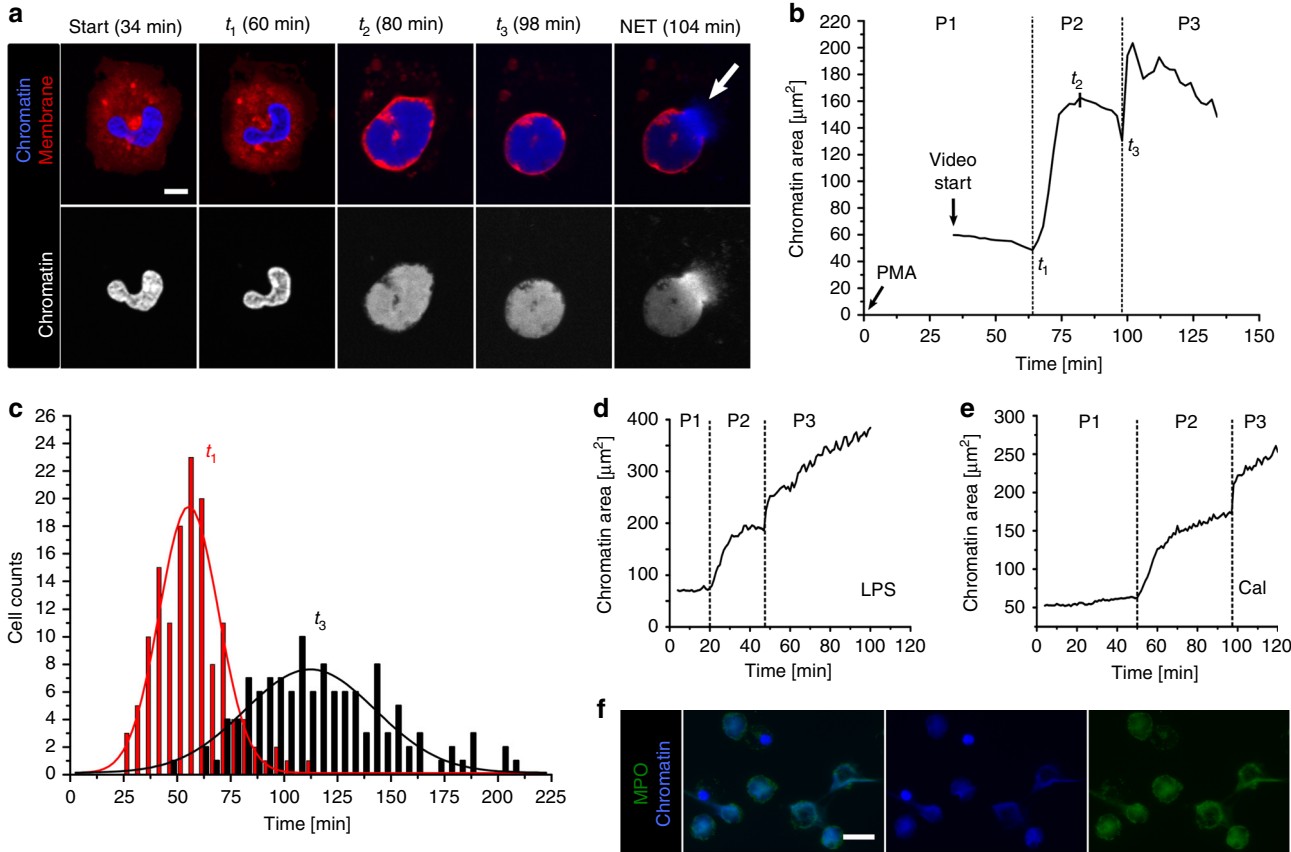

**Fig. 1** Phases of NETosis. **a** Morphological changes of chromatin (blue) and cell membrane (red) during NETosis of human neutrophils (stimulated with 100 nM PMA) imaged by live-cell confocal laser scanning microscopy (CLSM). The lobular nucleus loses its shape and chromatin decondenses until it fills the entire cell. Finally, the cell rounds up and releases the NET (white arrow). Scale bar = 5 µm. **b** Corresponding chromatin area of a NETotic neutrophil (**a**) as a function of time reveals three distinct phases. P1: Activation, lobulated nucleus. P2: Decondensation/expansion of chromatin within the cell ($t_1$ = start of chromatin expansion, $t_2$ = maximal chromatin expansion within the cell); cell rounding. P3: Rupture of the cell membrane (arrow **a**) and NET release ($t_3$ = NET release). **c** Histogram of onset times of the different phases. $n$ = 139 cells. $N$ = 5 donors. Lines represent Gaussian distribution function fits. **d** Time course of chromatin area for stimulation with LPS (Lipopolysaccharide, from Pseudomonas aeruginosa, 25 µg ml$^{-1}$). **e** Time course of chromatin area for stimulation with calcium ionophore (4 µM). **c–e** data acquired with live-cell wide field fluorescence microscopy. **f** Colocalization of decondensed neutrophil chromatin (blue) and myeloperoxidase (green). Fixed cells imaged by wide field fluorescence microscopy. Scale bar = 20 µm

of chromatin area at $t_1$ in Supplementary Fig. 5). Similar fluorescent labeling methods have been used by others to quantify nuclear envelope rupture events[32]. Previous publications have described the modification of nuclear lamins by phosphorylation as an early event, which would affect rigidity and could facilitate the here-described breakage of the nuclear envelope[33]. It should be noted that the breakage of the nuclear envelope appears to be a distinct process from the previously described dissolution of the nuclear envelope, which is a hallmark of late stages of NETosis[13,33].

In line with these previously published observations, we could show that the nuclear envelope further decomposed during P2 and P3 and lamin B1 was found distributed throughout the cytoplasm (Fig. 2c, P2/3). Subsequently, nuclear envelope breakdown allowed further expansion and swelling of chromatin within the cell (Fig. 2d). Consequently, the temporal correlation between $t_1$ and the rupturing of the nuclear envelope indicates that chromatin swelling is the physical driving force of this event.

The dissection of NETosis into distinct phases allowed us to identify and distinguish active (biological/biochemical) and passive (material properties) events. In the next step, we linked the phase classification to biochemical processes.

**Active and passive mechanisms during NETosis.** The initial steps of NETosis are thought to rely on several enzymes, with the exact progression depending greatly on the activator used to initiate NET formation[28]. In most scenarios, NETosis depends on the activity of typical neutrophil enzymes such as neutrophil lactase (NE) and myeloperoxidase (MPO)[34], as well as histone citrullination by the enzyme peptidyl-arginine deiminase 4 (PAD4). However, it is unclear at which time point during NETosis signaling and activity by these players are essential, and whether they initiate or maintain the process. If they were required in P1, and later phases were governed by passive mechanisms such as swelling of chromatin, then P2 and P3 should not depend on an active cellular energy supply. Indeed, ATP levels of (PMA) activated neutrophils quickly decreased in P1 by up to 70%, particularly within the first 30 min, indicating energy-dependent processes (Fig. 3a). In contrast, ATP levels then remained constant throughout P2 on a low level (Fig. 3a). To corroborate the hypothesis that energy supply is not necessary for P2, the main energy source in neutrophils, glycolysis[35,36], was cut off by inhibiting glucose metabolism with 2-Deoxy-D-glucose (2-Deox-Gluc)[37–40], which quickly and durably reduced ATP levels of neutrophils as early as 15 min in PMA-stimulated cells

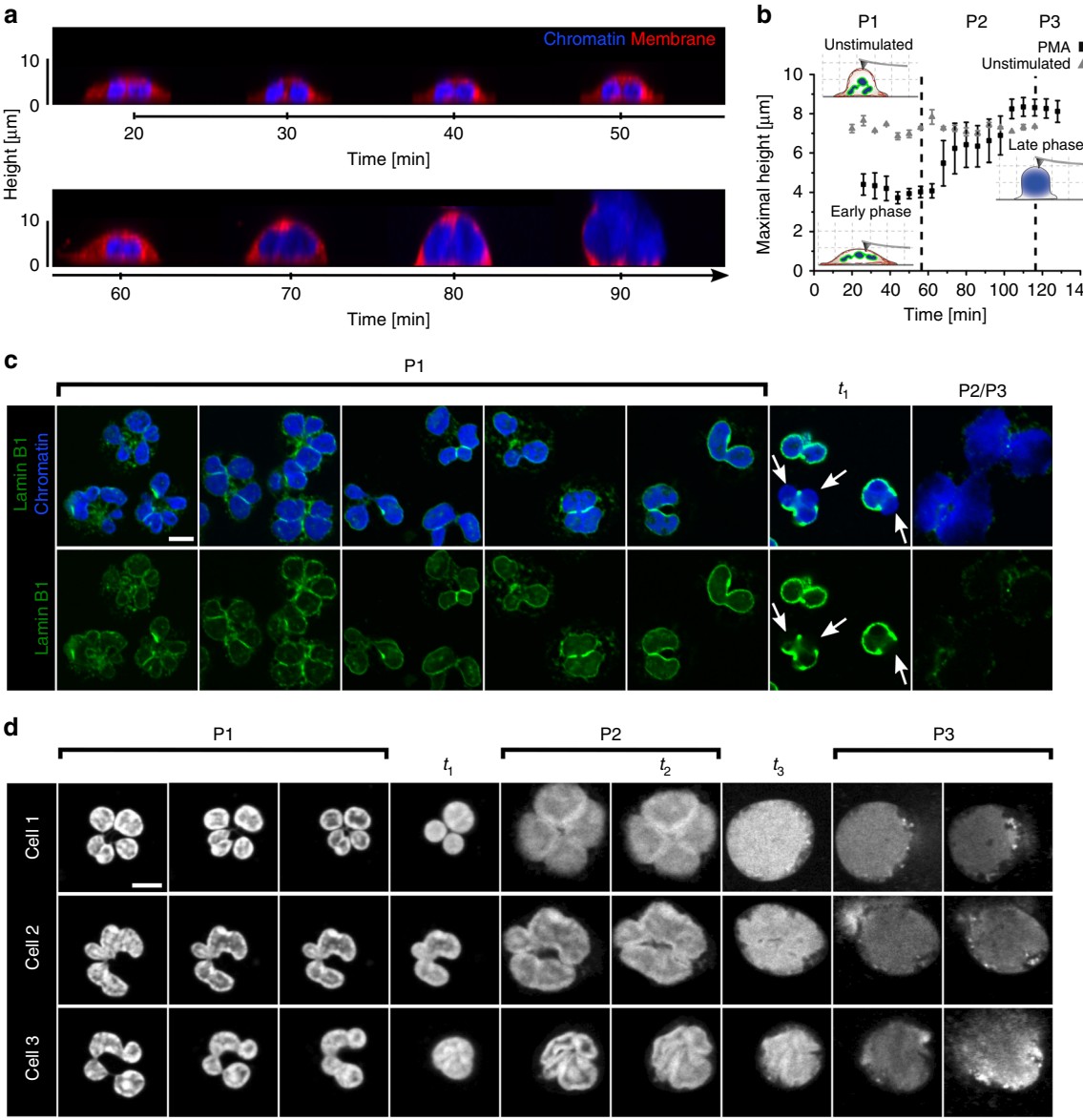

**Fig. 2** Chromatin swelling drives morphological changes. **a** Live-cell CLSM side view of a neutrophil during NETosis. Chromatin (blue) decondenses/ expands, reaches the membrane (red) and the cell rounds up until the membrane ruptures. z-stack depth: 1 μm. **b** Cell height as measured by atomic force microscopy (AFM) on life neutrophils. PMA stimulated cells adhere and flatten (compared to the control cells that stay more or less round) and then round up (>8 μm) in P2. $n = 3$. Mean ± SEM. **c** Characteristic distribution of lamin B1 (green) in the three phases, CSLM images of fixed cells. Lamin B1 first surrounds single lobuli of the nucleus/chromatin (blue). When chromatin starts to expand corresponding to the start of P2 (around $t_1$), the lamin B1 layer/ nuclear envelope ruptures on at least on one side of the nucleus. During P2 and P3 lamin B1 further decomposes. White arrows indicate rupture sites of the lamin B1 layer. Scale bar = 5 μm. **d** The original shape of the nucleus remains recognizable during the expansion process, particularly in the first part of P2 ($t_1$ to $t_2$), indicating isomorphic chromatin swelling and not directional transport (Supplementary Movies 13–15). In P1 the nucleus has a lobulated structure, which is maintained (self-similarity) during P2. Finally, the membrane is reached and, for a short period of time, this barrier prevents further expansion until it burst. Scale bar = 5 μm. Live-cell CSLM images

and within 60 min in unstimulated neutrophils (Supplementary Fig. 6b, c). Additionally, sodium azide (NaN₃) was used in this setup as a general inhibitor of metabolic function and, specifically, of metalloproteases[41–43], and MPO was inhibited with 4-aminobenzoic acid hydrazide (4-ABAH)[44,45]. We verified the function of these enzymatic inhibitors by showing that NaN₃ inhibits ROS production in neutrophils immediately after addition for at least 30 min (Supplementary Fig. 6a and 9c) and that 4-ABAH inhibited purified MPO within 1 min and stable within 15 min after PMA activation (Supplementary Fig. 6d). Thus, 4-ABAH directly interferes with PMA-induced NET formation[5]. None of the here-used inhibitors had any measurable effect on

NET-production of naïve neutrophils (Fig. 3b), nor did any of them show significant toxicity (Supplementary Fig. 7a). As expected, however, effects of these inhibitors were not exclusive to NET-formation, as 4-ABAH and 2-Deox-Gluc clearly decreased the uptake of FITC-labeled E. coli particles, although NaN₃ showed no effect in this setup (Supplementary Fig. 7b).

When added directly after stimulation all inhibitors significantly reduced NET formation (Fig. 3b). This effect successively decreased when inhibitors were added at later time points. Exposure to NaN₃ after >60 min and to 2-Deox-Gluc or 4-ABAH after >75 min no longer affected the number of decondensed nuclei. This result again implies that P1 depends on energy supply

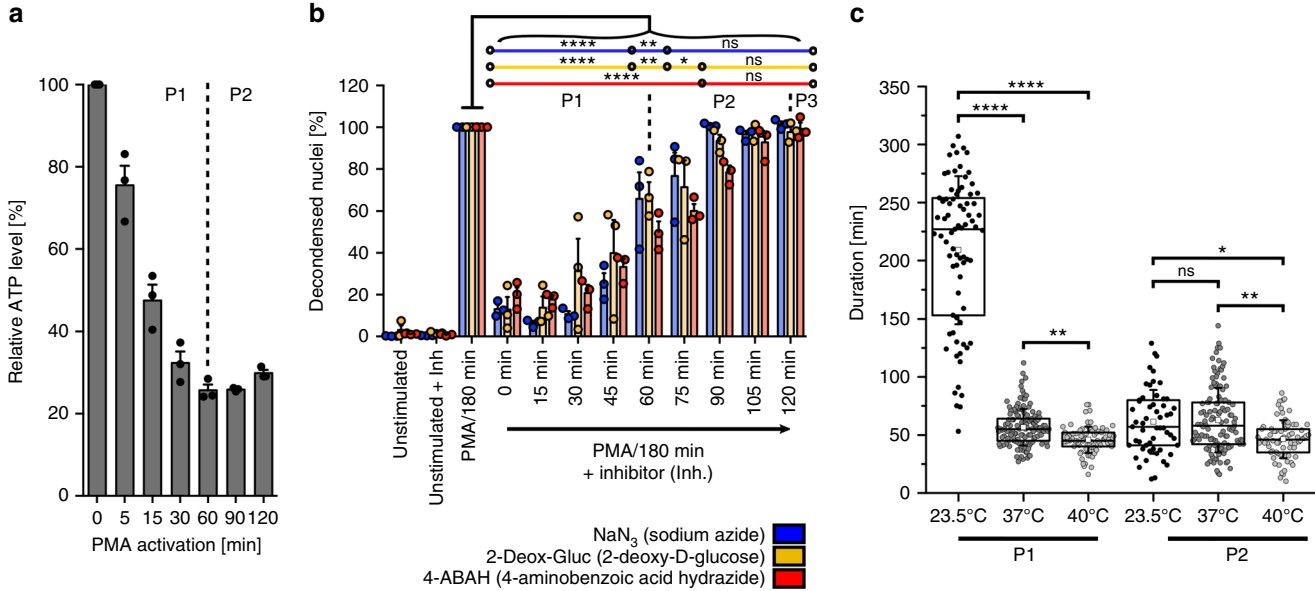

**Fig. 3** Active and passive processes during NETosis. **a** ATP levels in stimulated neutrophils decrease during P1 and reach a plateau in P2. Inhibition of glucose metabolism further reduces ATP levels (Supplementary Fig. 6c). $N = 3$. Mean ± SEM. **b** Metabolic inhibitors (sodium azide/3 mM, 2-deoxy-D-glucose/5 mM, 4-aminobenzoic acid hydrazide/100 µM) influence NET formation determined as relative number of decondensed nuclei after 180 min compared to activation with PMA only. All inhibitors decrease NET formation when added in P1, while P2 is not or only slightly affected, indicating a point of no return. $N = 3$ donors. Statistics: two-way ANOVA (Bonferroni's multiple comparisons test; *$p < 0.05$; **$p < 0.01$; ****$p < 0.0001$; ns = not significant). Mean ± SEM. **c** Phase duration at different temperatures (23.5, 37, 40 °C). P1 is significantly prolonged at lower temperatures, whereas P2 displays no or marginal temperature dependence. $N = 3$ (23.5, 40 °C). $N = 5$ (37 °C). Statistics: Kruskal-Wallis test (Dunn's multiple comparisons test; *$p < 0.05$; **$p < 0.01$; ****$p < 0.0001$; ns = not significant). Life-cell imaging. Boxplots display the 25th and 75th percentile and the horizontal line the median. Hollow squares represent the mean and whiskers the SD

and enzymatic activity, while P2 and P3 do not. So far, NETosis has been generally considered as an active process that requires the aforementioned enzymes. Here, we show that this generalization does not hold true for the complete mechanism.

Another hallmark of enzymatic activity is temperature-dependence. We chose to show the impact of temperature variations on the impact of NETosis as a complementary, inhibitor-free approach to investigate the importance on enzymatic activity. We quantified the duration of the different phases of NETosis at physiological core temperature (37 °C), hypothermia (23.5 °C) and hyperthermia/fever (40 °C). Higher temperatures significantly accelerated P1 whereas P2 showed no or only a slight temperature dependence (Fig. 3c, Supplementary Movie 12), indicating high enzyme activity in P1. If one assumes Arrhenius-like kinetics $\left(k \sim \exp\left(-\frac{E_a}{k_B T}\right)\right)$, the 4.14-fold shortened duration of P1 (227.5 min at 23.5 °C vs. 55.0 min at 37 °C) corresponds to an activation energy of around 80 kJ mol$^{-1}$, which falls into the range expected for enzyme-catalyzed reactions[46], and again corroborates our hypothesis of a switch from biochemically driven processes to behavior governed by material properties. P3 was not evaluated in the context of temperature-dependency as it is not possible to determine an end-point of P3 after the release of the NET.

Additionally, this result indicates an enzyme activity independent diffusive process in P2 since one expects lower temperature dependence for diffusion. In the first part of P2 (Fig. 1b, 2d, Supplementary Fig. 3a, Supplementary Movies 13–15) the chromatin area $A$ increased linearly with time $t$ and can be interpreted as a 2D diffusion process ($A \approx \langle x^2 \rangle = 4Dt$). The corresponding effective diffusion constant $D$ of 0.0108 µm$^2$ s$^{-1}$ at 37 °C (Supplementary Fig. 8a) is roughly in agreement with the diffusion of a $2 \times 10^9$ DNA sequence of ($D \approx 0.002$ µm$^2$ s$^{-1}$, $T = 37$ °C)[47,48].

**Entropic chromatin swelling drives morphological changes**. At the beginning of NETosis (P1 in our classification), histones are modified chemically (decrease of positive charge) by enzymes such as PAD4 or NE, which reduces the counterforces that hold the negatively charged DNA/chromatin together[10,49]. A condensed nucleus is under considerable entropic pressure as the radius of gyration of the human genome (length around 2 m) is approximately 150–200 µm[50,51]. Once the counterforces are no longer high enough to balance the entropic pressure, chromatin begins to swell. This time-point corresponds to $t_1$ (onset of P2) and marks a point of no return.

Another line of evidence pointing to entropic pressure as a relevant factor stems from the observation that small neutrophils rupture faster than larger ones after chromatin filled the whole cell lumen (Fig. 4a). As all cells contain the same amount of chromatin, entropic pressure exerted on the cell membrane is higher if they are smaller. This should lead to earlier rupturing of the membrane. Likewise, large intact neutrophils accumulate during the experiments because smaller ones rupture and release NETs first (Fig. 4b).

To analyze whether the swelling pressure generated during P2 determines if and when the membrane ruptures, we calculated the pressure exerted by chromatin and compared it to the rupture delay time ($t_2$ to $t_3$). For that purpose we modified a Navier-Stokes equation-based theory that describes pressure as a function of time $t$ (see Methods[52]):

$$p(t) = \frac{\eta(R(t))}{l_p^2(R(t))} \frac{dR(t)}{dt} R(t) \tag{1}$$

Here $\eta$ is the viscosity of the chromatin (liquid), $R(t)$ the effective radius of the chromatin area and $l_p$ the chromatin mesh size. STED nanoscopy images of chromatin just before the cells

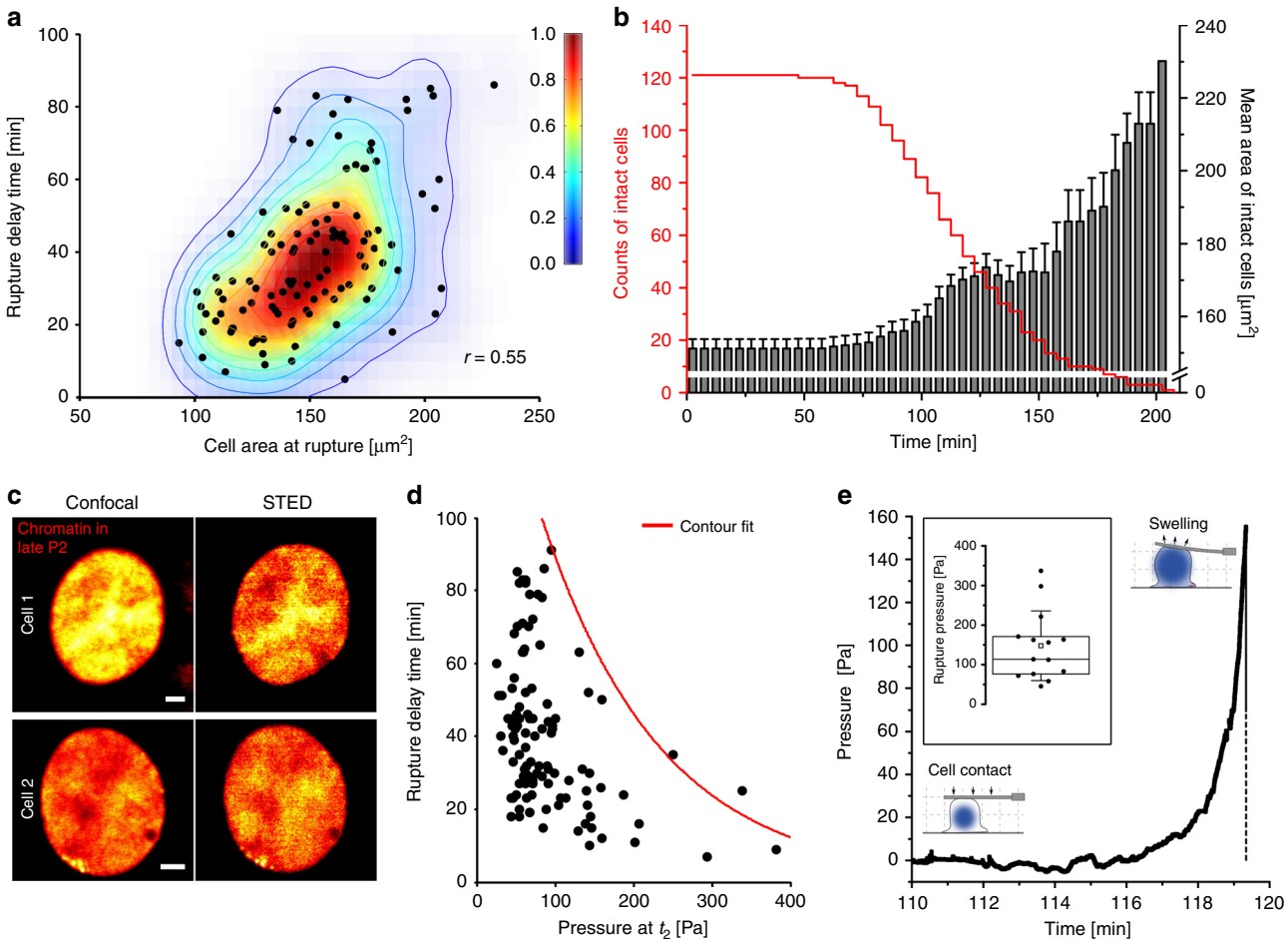

**Fig. 4** Entropic swelling of chromatin causes membrane rupture. **a** Correlation between cell area at $t_3$ (chromatin area at $t_3 \approx$ total cell area at $t_3$) and time span until membrane rupture occurs following maximal chromatin expansion ($t_3$– $t_2$, rupture delay time). Larger cells rupture later than smaller cells. $n =$ 112 cells (only cells of population 1 included, see Supplementary Fig. 3a, b). Fit lines show normalized 2D-probability density function calculated by kernel density estimation (KDE). $N =$ 5 donors. **b** Average cell area (at $t_3$) of remaining/intact cells. The area of intact cells increases from $\approx$151 to $\approx$230 $\mu m^2$ with time, indicating earlier rupture of small cells. $n =$ 121 cells. $N =$ 5 donors. Mean ± SEM. **c** Live-cell confocal (left row) and STED (right row) images of chromatin (SiR-DNA) of two neutrophils undergoing NETosis (late P2) show almost uniform distribution of chromatin suggesting a mesh size smaller than the resolution of the microscope ($\approx$120 nm (xy) and $\approx$150 nm (z), Supplementary Fig. 8b). Scale = 2 $\mu m$. **d** Time delay until rupture as a function of the calculated pressure $p$ at $t_2$. $n =$ 112 cells (only cells of population 1 included, see Supplementary Fig. 3a, b). $N =$ 5 donors. The contour fit is generated by fitting the maximal data points to an exponential decay curve. **e** Neutrophils undergoing NETosis exert an increasing pressure on a fixed AFM cantilever until they rupture (end point of measurement). Inset: $N =$ 5 ($n =$ 14 cells). Boxplots display the 25th and 75th percentile and the horizontal line the median. Hollow squares represent the mean and whiskers the SD

ruptured showed no fine structure indicating that the mesh size is below the resolution of this microscope (about $\approx$120 nm in xy-direction and $\approx$150 nm in z-direction) (Fig. 4c, Supplementary Fig. 8b, c). Therefore, we assumed that the whole genome (around 2 m DNA) is evenly arranged inside the cell and estimated that $l_p$ to be around 20 nm. Cells exposed to higher pressure ruptured faster than their smaller counterparts (Fig. 4d). The calculated pressure values are in a similar range as pressure values known from osmotic lysis experiments of lipid vesicles[53]. To find out if these calculated pressure values were actually exerted by the cell we measured these forces directly with AFM (Fig. 4e, Supplementary Fig. 8d). A cantilever was positioned in direct contact with the cell and deflected by the rounding cell. This is a known approach to measure rounding forces e.g. during mitosis[54]. The pressure exerted by the cell increased with time and lifted the cantilever until the cell finally ruptured (end point of measurement). The measured pressure values between 100–200 Pa are in the same range as the calculated pressure values from Fig. 4d. Together with the control experiments that show that energy

supply (Fig. 3) and the actin cytoskeleton (Fig. 5, below) are not crucial for P2 this result further supports chromatin swelling as the major driving force in P2.

In conclusion, our results show that P1 of NETosis is a biologically active process governed by signaling and enzymatic reactions. In contrast, P2 is mainly passive, based on entropic chromatin swelling and cannot be stopped once it has started. Therefore, the onset of P2 ($t = t_1$) represents a point of no return.

**Impact of cytoskeletal rearrangements on NETosis.** In addition to the unexpected importance of chromatin dynamics reported this far, the cytoskeleton may also contribute to shape shifts and DNA expulsion. While cytoskeletal components appear to rearrange and F-actin may become degraded during NETosis[4,34], these processes remain poorly understood. Figure 5a shows the F-actin distribution (labeled with SiR-actin) at different time points after PMA-stimulation. During the first 30 min F-actin became more prominent as expected for stimulation with the protein-kinase C activator PMA, which has multiple effects on

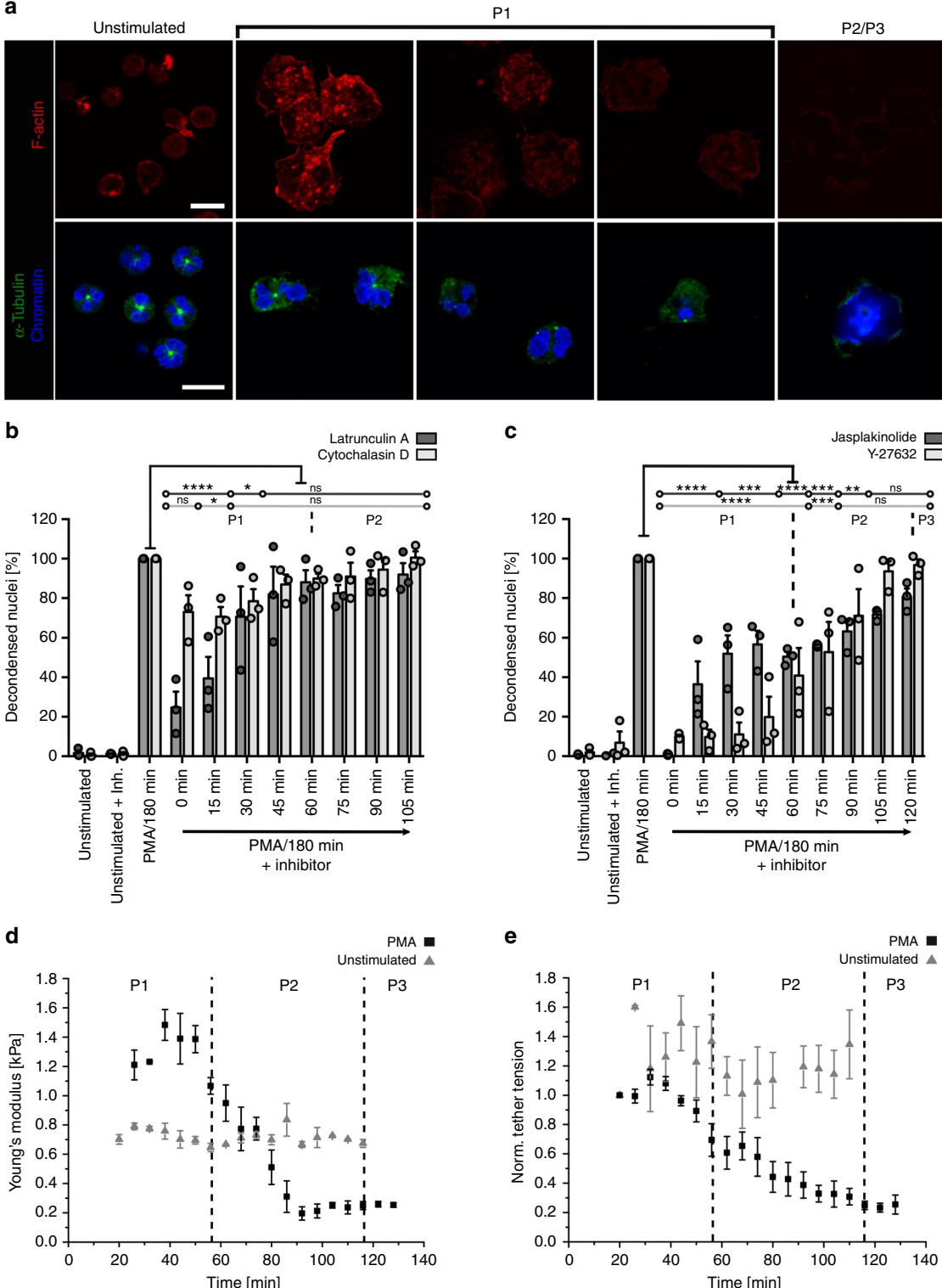

neutrophils, among them the induction of chemotaxis[55]. Then, it diminished and completely disassembled after 90–180 min (see quantification in Supplementary Fig. 9a).

As recently described, α-tubulin is initially organized in filaments originating from the microtubule organization center (MTOC; unstimulated cells)[33]. One could speculate that these MTOCs are involved in active transport of chromatin. However, during phase I after PMA-stimulation, α-tubulin filaments disappeared and were rearranged into dot-like structures,

reminiscent of mitotic centrosomes. These were visible until the beginning of chromatin decondensation, namely until the beginning of phase 2 (Fig. 5a). As we did not observe these filaments any more during phase 2, it is unlikely that active chromatin transport along these centrosomes like structures plays a role during the expansion of chromatin.

Additionally, we sought to determine the role of the cytoskeleton through specific inhibition: Jasplakinolide (10 µM, actin stabilization/induction of actin polymerization),

**Fig. 5** Rearrangement of the cytoskeleton and evolution of mechanical properties. **a** At the beginning of NETosis F-actin (red) is laterally enriched and localizes in the lamellipodia. α-Tubulin filaments (green) are arranged originating from the microtubule organizing center (MTOC) in unstimulated cells. Within the next hours (during P1) cytoskeletal components disintegrate. Remaining F-actin accumulates at the cell margin and α-tubulin is first rearranged in centrosome-like structures which disappear at the beginning of P2. CLSM images of fixed cells. Activation = PMA (100 nM). Blue = chromatin. Scale = 10 μm. **b**, **c** Inhibition of NET formation with the F-actin polymerization inhibitors Cytochalasin D (100 nM) and Latrunculin A (1 μM), F-actin-stabilizing drug Jasplakinolide (10 μM) and the ROCK-inhibitor Y-27632 (19.2 μM) significantly reduces the formation of NETs (measured as %-relative number of decondensed nuclei after 180 min compared to activation with PMA only) in P1, while P2 depends less or not on F-actin stabilization and ROCK-inhibition. Statistics: two-way ANOVA (Bonferroni's multiple comparisons test; *$p < 0.05$; **$p < 0.01$; ***$p < 0.001$; ****$p < 0.0001$; ns = not significant). $N = 3$ donors. Mean ± SEM. **d** Normalized tether tension of life neutrophils (measured with AFM) decreases over the entire time course (raw data: >0.35 mN m to <0.07 mN m) of PMA-activated NETosis indicating a loss of cytoskeletal stability. Values of control cells remain stable. $N = 3$. Mean ± SEM. **e** Cell stiffness (Young's modulus) of life neutrophils decreases from >1.5 kPa to <0.3 kPa after stimulation with PMA whereas the stiffness of control cells remains constant. $N = 3$. Mean ± SEM

Cytochalasin D (100 nM, inhibition of actin polymerization), Latrunculin A (1 μM, inhibition of actin polymerization) and Docetaxel (100 nM, inhibition of tubulin depolymerization) were added at different time points after stimulation (with 100 nM, PMA). While Docetaxel exerted no (Supplementary Fig. 9b), Cytochalasin D and Latrunculin A showed a significant reduction of NETosis when added up to 15 min after activation (Fig. 5b). Jasplakinolide inhibited NETosis completely when added from the beginning. The latter effect decreased as NETosis advanced (Fig. 5c). Based on this finding we concluded that reorganization of the cytoskeleton is not only a consequence of NETosis, but also a requirement during the active phase (P1; Fig. 5b, c). Additionally, we manipulated cell motility by inhibiting the rho-associated coiled-coil-containing protein kinase 1 and 2 (ROCK 1/2) signaling pathway using Y-27632 (19.2 μM). Again, we found a strong influence on P1 (Fig. 5c). These results are in good agreement with our results on enzyme activity (Fig. 3b). While it has been shown that cytoskeletal inhibitors may influence the NADPH oxidase[56], we could rule out that these off-target effects were causing the reduction in NETosis, as Jasplakinolide inhibited ROS formation, while Latrunculin A increased it and the Y-27632 and Cytochalasin D had no or only a slight effect on ROS production (Supplementary Fig. 9c). Thus, while effects on ROS production were very heterogeneous, the effect of cytoskeletal inhibition on NET production showed the same pattern for all used inhibitors (Fig. 5b, c). Therefore, we can rule out that the observed effects on NETosis are exclusively mediated by an influence on the NADPH oxidase. None of our cytoskeletal inhibitors produced any significant cytotoxic effect on neutrophils (Supplementary Fig. 7a).

In conclusion, NETosis requires an intact and functional actin cytoskeleton at the beginning of P1 and the reorganization of F-actin is essential to proceed to P2 and P3. Rearrangement of the microtubule apparatus appears to be a marker of the activation of biochemical pathways required for NETosis (namely the activation of cyclin-dependent kinases 6 and 4)[33] and an influence of certain microtubule inhibitors has been postulated by isolated publications[57]. However, in our setup the microtubule apparatus itself appears to be dispensable for NET formation. Taken together, both the actin cytoskeleton of the neutrophils, as well as the microtubule apparatus become dysfunctional as NETosis progresses. Therefore, it is unlikely that the cytoskeleton plays an active role during the final steps of NETosis. Dissolution of these major components, which normally stabilize the cell, likely impairs the cell's mechanical properties, thus helping the final cell membrane rupture.

To test this hypothesis, the cells' mechanical properties were measured by atomic force microscopy (AFM) (Fig. 5d, e, Supplementary Fig. 10, Supplementary Movie 16). Indeed, cells became markedly softer during NETosis, particularly in P2, as

evidenced by a decrease of the Young's modulus from $E = 1.5$ to 0.3 kPa (Fig. 5d).

Additionally, retraction curves of individual membrane tethers were analyzed to distinguish the cell's bulk mechanical properties from the mechanical properties of the membrane alone. The membrane tension decreased by 77% (0.35 to 0.07 mN m$^{-1}$) (Fig. 5e). Although alterations of the plasma membrane are likely to occur in an inflammasome-dependent manner during NETosis[58], such a dramatic decrease cannot be explained by changes of membrane composition only, but must be caused by the disassembly of the actin cortex beneath the membrane[59,60]. The membrane tension $T$ in late P2 corresponds to a membrane pressure $p \approx \frac{2}{R} T \approx 20$ Pa (Young-Laplace equation, radius $R \approx$ μm). The calculated swelling pressure (Fig. 4d) is in the same range indicating that chromatin swelling is sufficient to break the membrane.

**Location of membrane rupture**. To test the hypothesis that the chromatin swelling pressure determines the membrane breaking point, we correlated swelling speed and anisotropy with the position of the rupture point. Even though swelling itself should be isotropic, the position of the nucleus/lobuli inside the cell varied with respect to the cell boundary. We first looked at chromatin decondensation and calculated the average swelling speed during $t_1 < t < t_2$ (represented by a velocity field, Fig. 6a, Supplementary Fig. 11). Interestingly, in cells in which the nucleus was closer to one side of the cell membrane, the rupture point was usually close to areas of low chromatin movement, e.g., in close proximity to the cell membrane (cell 1–3). In these cases, there was less space for chromatin to expand (higher remaining entropic pressure). If (in the minority of cells) the nucleus was centered in the cell the rupture direction was random (cell 4).

Moreover, cells typically ruptured at the side that experienced the last membrane retraction close to the long axis of the cells before they rounded up (Fig. 6b).

In order to quantify these observations, a rupture point axis was determined by connecting the rupture point A with the center of mass of the respective cell ($M$). Then, the cells' outlines were fitted with an ellipse and the cell membrane shrinking velocities of both cell sides (A and B) were analyzed along the rupture axis ($v_A$ and $v_B$) during the last minutes before rupture (30 min ± 16 min) (Fig. 6c). Cells ruptured at the side that experienced significantly more movement after the rounding process started in late P2 ($t > t_2$) (Fig. 6d). Most rupture events occurred close to the previously long axis of the (elliptic) cell (Fig. 6e), thus allowing the prediction of the NET release location.

This analysis of membrane morphology shows that cells become circular before the membrane ruptures on the last moving side close to the previous long axis. Additionally, the position of the nucleus with respect to the membrane determines

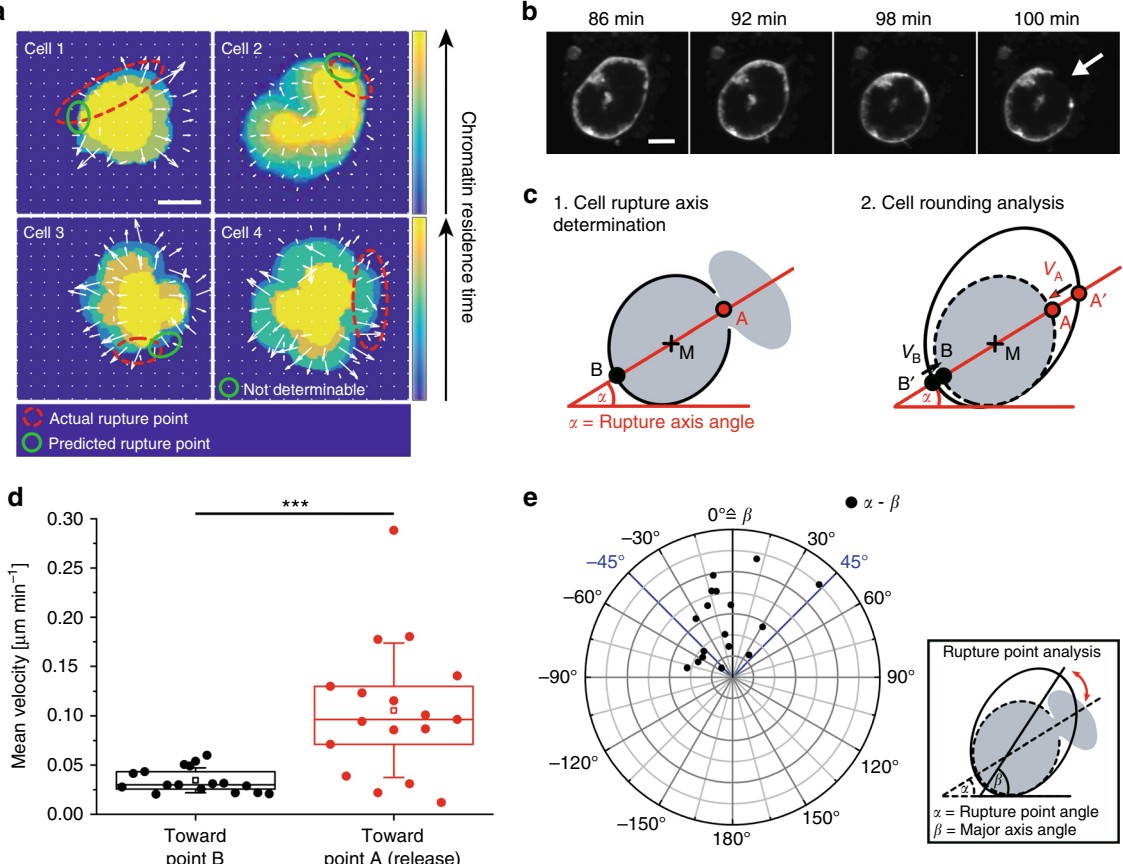

**Fig. 6** Predetermination of the membrane rupture point. **a** Velocity plots of chromatin swelling. Changes to darker colors indicate faster movement (shorter residence time) of chromatin. The actual rupture point (red circle) often correlates with areas of slow movement (predicted rupture point, green circle). **b** Live-cell CLSM images of the cell membrane (PKH26 staining) directly before NET release. The cell rounds up and ruptures when maximum circularity is reached ($t = 98$ min). Scale bar = 5 μm. **c** Schematic of the rupture point analysis. Rupture points were analyzed by (1) fitting an ellipse to the cell before it became round and determining the rupture axis between the rupture point A and the center of mass M and (2) determining the retraction speed ($v_A$ and $v_B$) on both sides of the (previously elliptic) cell (see Methods). **d** Shrinking velocity of the two opposing cell poles (A and B). The neutrophil retracts its membrane with a significantly higher velocity at the future rupture site (A). $n = 17$. $N = 4$ independent experiments. Statistics: Mann–Whitney test, two-tailed ($***p < 0.001$). Boxplots display the 25th and 75th percentile and the horizontal line the median. Hollow squares represent the mean and whiskers the SD. **e** Angle plot shows that the membrane ruptures in proximity of the major axis. $\alpha$ = rupture point angle. $\beta$ = major axis angle. $n = 17$. $N = 4$ independent experiments

on which side of the cell entropic chromatin pressure becomes higher.

An additional aspect that has not been addressed in detail so far is influence of adhesion in NETosis. We used reflection interference contrast microscopy (RICM) (Supplementary Fig. 4c, d, Supplementary Movies 9–11) to image the interface between cell and substrate and address this question[61,62]. Neutrophils quickly and strongly adhered and left membrane behind, while rounding up. To further test the impact of adhesion we quantified NETosis on differently coated surfaces (Supplementary Fig. 12). Interestingly, on surfaces passivated with poly(L-lysine)-graft-poly(ethylene glycol) (PLL-g-PEG) neutrophils still performed NETosis even though they could not properly adhere (Supplementary Movie 17). This result does not rule out an important role of adhesion in determining the threshold for NETosis especially for different (weaker) activators. However, it shows that once the cell initiated NETosis, additional adhesive cues were not important anymore. Nevertheless, the influence of external factors for the initiation and execution of NETosis, including adhesive cues and surface characteristics like surface stiffness, for example, certainly warrant further investigation in the future.

## Discussion

Over the last years, much effort has been put into unraveling the signaling cascades and enzymatic players that are indispensable for NETosis. Here, we provide a comprehensive and unique picture of the complex biophysical aspects that govern the different phases of NETosis using an interdisciplinary approach and innovative imaging methods. We conclude that NETosis is a highly organized process with a first phase (P1) that is governed by biochemical modifications including histone citrullination and phosphorylation of lamins[33] that prepare the cell for later mechanical changes. We show that a point of no return exists after which active processes such as enzymatic activities become secondary and the cells behavior is determined by the characteristics of chromatin (Fig. 7, Supplementary Fig. 13). Morphological changes, as well as rupture/burst of nuclear envelope and cell membrane are driven by entropic swelling of chromatin. At this point, pharmaceutical inhibition of NETosis is no longer possible. These findings may also prove important for other biological processes, such as cell division or other forms of cell death[54,63]. Indeed, it has been shown very recently that broad overlaps exist between NETosis and mitosis from a biochemical

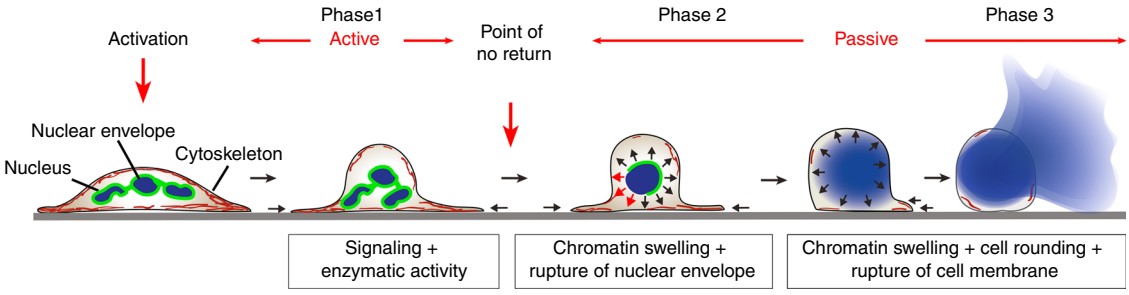

**Fig. 7** Biophysical model of NET release NETosis can be divided into three distinct phases (according to chromatin status) that are separated by a point of no return. The major physical driving force for morphological changes and NET release after phase 1 is entropic swelling of chromatin

point of view. It will be worthwhile studying these similarities from a biophysical perspective[33].

Ultimately, cells appear to be more than biochemical factories and complex processes such as NETosis can therefore be driven not only by biochemical signaling but also material properties.

Chromatin is traditionally viewed as a nuclear entity that mainly regulates gene expression. Here, we show that chromatin is much more than a template for information processing but is able to actively conduct biological processes.

## Methods

**Isolation of human neutrophils.** Neutrophils were isolated from human venous blood of healthy donors. The study was approved by the ethics committee of the university medical center goettingen (chairman Prof. Dr. med. Jürger Brockmöller) and fully informed consent of all donors obtained after clearing the possible consequences of the study. The isolation was performed according to previous published standard protocols.[1]

In short, fresh blood was collected with S-Monovettes KE 7.5 ml (Sarstedt). Blood was gently layered in a 1:1 ratio on top of Histopaque 1119 (Sigma-Aldrich) and centrifuged at $1100 \times g$ for 21 min. Then, the transparent third and pink fourth layer containing the white blood cells were collected and mixed with HBSS (without $Ca^{2+}/Mg^{2+}$, Thermo Fisher Scientific). Cells were pelleted by centrifugation for 10 min at $400 \times g$. After discarding the supernatant, the pellet was resuspended in HBSS without $Ca^{2+}/Mg^{2+}$ and layered on top of a phosphate buffered percoll (GE Healthcare) gradient with the concentrations 85, 80, 75, 70, and 65% and centrifuged at $1100 \times g$ for 22 min. The accumulated neutrophils were received by collecting half of the 70%, full 75% and half of the 80% layer and washed with HBSS. The remaining cell pellet was resuspended in 1 ml HBSS. Cells were counted and suspended at the required concentration for the following procedures with RPMI 1640 (Lonza) containing 10 mM HEPES (Roth) and 0.5% human serum albumin (HSA) (Sigma-Aldrich). Experiments with lipopolysaccharides (LPS from Pseudomonas aeruginosa serotype 10.22, strain: ATCC 27316, Sigma-Aldrich) or calcium ionophores (CaI, Sigma-Aldrich) were carried out without addition of 0.5% human serum albumin. Cellular identity was confirmed by a cytospin assay (Cytospin 2 Zentrifuge, Shanson) followed by Diff Quick staining (Medion Diagnostics). Cell purity was >95% of isolated cells (without erythrocytes).

**Live cell imaging (fluorescence microscopy).** Fresh isolated human neutrophils were seeded ($4–5 \times 10^5$ cells per ml) on ibidi treat flow chambers (μ-Slide I 0.8 Luer ibidi Treat, ibidi GmbH) for 30 min (37 °C, 5% $CO_2$) and stained with 1.62 μM Hoechst 33342 (Sigma-aldrich) for 15 min (37 °C, 5% $CO_2$). For membrane staining, cells were stained before seeding with 2 μM PKH26 (PKH26-kit, Sigma-aldrich) following the companies' instructions. Cells were activated for NETosis with 100 nM Phorbol-12-myristate-13-acetate (PMA, Sigma-aldrich), 4 μM CaI or 25 μg ml$^{-1}$ LPS. Live cell imaging was performed at 23.5, 37 or 40 °C (ibidi heating system, ibidi GmbH) for 3–5 h with minimized light exposure. Bright field microscopy movies were obtained ×16 magnified (EC Plan-Neofluar Ph1/440331-9901-000, Zeiss) using the camera CoolSNAP ES (Photometrics) and the microscope Axiovert 200 (software: Metamorph 6.3r2., Molecular Devices, Zeiss) with a frame rate of one picture per minute in the blue channel (Filter set 02 shift free/ 488002-9901-000, Zeiss). For CLSM images an Olympus IX83 inverted microscope (software: Olympus Fluoview Ver.4.2, Olympus) was used and the movies were recorded ×60 magnified (UPlanSApo 1.35 oil, Olympus). Hoechst fluorescence was detected at 405 nm and PKH26 fluorescence at 561 nm. All 2D-movies were obtained with a frame-rate of one picture per 2 min and the 3D-movies with one picture per ten minutes and a z-stack depth of 1 μm per slice. All videos and pictures were further processed with ImageJ (v. 1.46r ad 1.50c4; National Institutes of Health) and MATLAB (v. R2008a/R2014a; The MathWorks, Inc.) as described in the section statistics and data analysis.

**Inhibitor experiments.** Fresh isolated human neutrophils (10,000 per well) were seeded in 96-glassbottom-well-plates (In vitro scientific) and activated for NETosis with PMA, final concentration 100 nM. Subsequently, the function of cytoskeletal components was inhibited with Cytochalasin D (100 nM, Abcam), Latrunculin A (1 μM, Sigma-Aldrich), Docetaxel (100 nM, Abcam), Jasplakinolide (10 μM, Enzo) or Y-27632 (19.2 μM, Abcam) and enzyme activity inhibited with 2-deoxy-D-glycosis (2- Deox-Gluc, 5 mM, Sigma-aldrich), sodium azide ($NaN_3$, 3 mM, Merck) or 4-Aminobenzoic acid hydrazide (ABAH, 100 μM, Cayman) at defined time points (0 min, 15 min, 30 min, 45 min, 60 min, 75 min, 90 min, 105 min, and 120 min) after activation. All experiments were performed in triplicates. To stop NET formation, cells were fixed with 2% PFA final concentration (Roth) after 3 h incubation (37 °C, 5% $CO_2$) and stored over night at 4 °C. The fixed probes were washed 10 min with 1× PBS (Lonza) and chromatin stained with Hoechst at room temperature. After staining, cells were washed with PBS and imaged with the microscope Axiovert 200 (×16 magnification, Zeiss; software: Metamorph 6.3r2, Molecular Devices) and a CoolSNAP ES camera (Photometrics) in the blue channel (Filter set49 DAPI shift free, 488049-9901-000, Zeiss). For each well in total 5-6 images of different regions were collected. For all experiments, the amount of decondensed nuclei, as well as the total cell count was quantified (blinded) with ImageJ. Percentages of decondensed nuclei/NETs were calculated relative to the amount of decondensed nuclei/NETs after stimulation of cells with PMA for 3 h.

**ATP measurements.** Fresh isolated human neutrophils (10,000 per well in RPMI (10 mM HEPES, 0.5% HSA without phenolred)) were seeded in white 96-well-plates (Greiner bio-one) and activated with PMA in a final concentration of 100 nM for defined time points (5 min, 15 min, 30 min, 60 min, 90 min, and 120 min). After incubation, CellTiter-Glo® Reagent (Promega) was added in a 1:1 ratio and the ATP amount was determined following the company instructions. In short, the mixture was shaken for 2 min to induce cell lysis and incubated for 10 min at room temperature to stabilize the luminescence signal. Subsequently, luminescence was measured (GLOMAX® 96 Microplate Luminometer, Software: GLOMAX 1.9.3, Turner BioSystems) and the ATP levels were calculated relatively to the ATP amount of unstimulated cells incubated for 120 min.

**Staining procedures.** Fresh isolated human neutrophils (200,000 per well) were seeded on pretreated (99% alcohol) glass cover slips (#1.5) in 24-well plates (Greiner bio-one) and NET formation induced with 100 nM PMA. Cells were then fixed at different time points after NET formation with 2% PFA and stored in PBS over night. The following staining procedure was carried out based on previous published protocols[64]. Briefly, cover slips were gently removed from the 24-well plate and layered upside down on the washing solution (1xPBS). Cells were then permeabilized with a 0.1% TritonX (Merck) containing solution for 10 min at 4 °C, washed and blocked with 5% fetal calf serum (FCS, Merck) or 3% BSA (Lamin B1 staining). Subsequently cells were stained with monoclonal anti-human MPO (IgG, mouse, 1:500) (ab25989, Abcam), monoclonal anti-human α-Tubulin (IgG, rabbit, 1:50) (#2125, Cell Signaling Technology) or polyclonal anti-human lamin B1 (IgG, rabbit, 1:1000) (ab16048, Abcam) as primary antibodies over night (4 °C) and visualized with polyclonal anti-mouse Alexa488 (IgG, goat, 1:1000) (#4408, Cell Signaling Technology) or polyclonal anti-rabbit Alexa488 (IgG, goat, 1:500) (A-11034, ThermoFisher Scientific) as secondary antibodies. In case of staining with SiR-dyes, cells were not permeabilized but directly stained after washing with SiR-Actin (SC001, Spirochrome AG/Tebu-bio) or SiR-DNA (SC007, Spirochrome AG/Tebu-bio) at 3 μM. Then, chromatin was stained with Hoechst if applicable and cover slips were mounted with Faramount Mounting Medium (Dako Agilent Technologies) on microscopy slide. After complete drying and fixation with nail polish, samples were imaged with 40x magnification (Plan-Neofluar 40×/1.30 oil Iris/4440456-0000-000, Zeiss) in a fluorescence microscope (AxioImager M1, Software: AxioVision Rel.4.7, Zeiss) or ×60 magnified with confocal laser scanning microscopy (Olympus IX83 inverted microscope, software: Olympus Fluoview v.4.2).

**3D-STED microscopy.** In conventional optical microscopy, the resolution is limited by diffraction to about half the wavelength of light ($\lambda/2 \approx 300$ nm) and

about the wavelength ($\lambda \approx 600$ nm) in the lateral and axial direction, respectively. To resolve smaller features samples were stained with SiR-DNA as described in the staining procedure section and were imaged with a super-resolution STED (Stimulated Emission Depletion) microscope[65]. For live cell imaging, cells (10,000 per well) were seeded for 30 min in 10-well CELLview™ glass slides (Greiner Bio-one) in RPMI (10 mM HEPES, 0.5% FCS without phenolred) and activated with 100 nM PMA and stained with 1 μM SiR-DNA. In a canonical STED microscope, the diffraction-limited excitation spot is superimposed with a red-shifted donut-shaped laser beam (STED beam) featuring a zero-intensity at its center. The STED beam has the ability to inhibit fluorescence from excited molecules. The higher the STED intensity, the more efficient this inhibitory effect is. As a consequence, fluorescence is confined to a sub-diffraction sized area. The super-resolution image is recorded by scanning this area across the sample.

To assess information beyond the diffraction limit, experiments were performed with a custom-built 3D-STED microscope. The excitation beam of 640 nm wavelength is emitted from a Picosecond Pulsed Diode Laser Head (LDH-P-C-640B, PicoQuant, Berlin, Germany). As STED light source at 775 nm, sub-nanosecond laser (Katana-08, Onefive GmbH, Zurich, Switzerland) is used. To achieve a super-resolved image in 3D, the Easy3D-STED Module (Abberior Instruments GmbH, Göttingen, Germany) is used: a programmable spatial light modulator (SLM) creates the STED light phase patterns required for 3D STED microscopy. It allowed a lateral (xy) and axial (z) resolution enhancement with only one STED beam instead of two separate STED beams. The relative pulse delay between the excitation laser and the STED laser is set electronically. Laser focusing and fluorescence collection are performed by the same oil immersion objective (UPlanSApo 60 × 1.35 Oil, Olympus Corporation, Tokyo, Japan). An appropriate series of dichroic mirrors and optical filters separates the fluorescent signal from both, the excitation and the STED light, and sends it into the detector. Fluorescence photons are detected by a Single Photon Counting module (SPCM-AQRH-13-FC, Excelitas Technologies Corp., Waltham, MA).

Experiments were run with the software ImSpector (MPI für biophysikalische Chemie, Göttingen, Germany); data analysis was performed with ImageJ (U. S. National Institutes of Health, Bethesda, Maryland, USA, http://imagej.nih.gov/ij/).

**Atomic Force Microscopy (AFM).** For all AFM-experiments, 60,000 cells (62 cells per mm²) were seeded on an ibidi-treaded μ-dish (81156, Ibidi) for 30 min. Subsequently, the cells were activated with 100 nM PMA. For imaging, an Olympus IX81 microscope was used and steered with an Olympus CellSense Dimension software (v. 3.15). A custom-made heating system integrated into the dish holder enabled the temperature regulation of the sample ($T = 37$ C°).

For elasticity measurements, a non-conductive silicon nitride tip (MLCT, $f_0 = 10$–20 kHz, $k = 0.02$ N m$^{-1}$, Bruker) was directed by a JPK AFM-head system (JPK Instruments 00996, Nanowizard 3), calibrated according to the manufacturer's instructions (Nanowizard 3 user manual) and kept in the medium to equilibrate the temperature. The tip was slowly approached to the cell via force-feedback recognition and positioned above the middle of the cell. In general, the sample was measured over a 20 × 20 μm area with 8 × 8 force curves for each time point and controlled by a JPK SPM Control software (v.5). To prevent plastic deformation a relative set point of 0.5 nN was chosen together with an extension speed of 3 μm s$^{-1}$ and a delay time of 1 s between each force curve measurement resulting in a total iteration time of around 6 min. All force curves were manually reviewed, baseline corrected and the Young's modulus $E$ was calculated using the Hertz fit for pyramidal tip geometries

$$F = \frac{E}{1 - \nu^2} \frac{\tan(\alpha)}{\sqrt{2}} \delta^2 \qquad (2)$$

with a Poisson's ratio set to $\nu = 0.5$, a face angle of $\alpha = 20°$, the measured force $F$ and the tip-sample distance $\delta$ of the force curve. The mean Young's modulus was calculated from the force-curve in the middle of the cell and four direct neighbors to avoid edge effects.

Parallel imaging to verify NETosis and healthiness of the cells were performed on the same system. Here, cells were stained before with 1.62 μM Hoechst (Sigma-Aldrich) and observed with an Orca Flash 2.8 camera (C11440, Hamamatsu) at ×40 magnification (LUCPlan FLN, Olympus). For illumination, an integrated lamp system (cellTIRF-4Line System + IUX-C2, Olympus) was used and filtered by a DAPI filter set (#86-370-OLY, Olympus).

To measure tether tension during NETosis, the baselines of all AFM retraction curves on the cell were analyzed for step like deflection changes that indicate tether formation (sharp increase of cantilever deflection and constant variance before and after). All restoring forces were measured with a JPK Data Processing Software (V. spm 5.0.69, JPK Instruments) and stored separately for each frame time $t$. The membrane tension $T(t)$ was then calculated by using the relation[66–68]

$$T(t) = \frac{F(t)^2}{8B\pi^2} \qquad (3)$$

with $B$ representing the bending stiffness of a lipid membrane (set to $3 \times 10^{-12}$ dyne cm) and $F(t)$ the respective tether force[2]. For each frame, these values were averaged ($n \approx 3$) and each data set was normalized to its first value $T(t_0)$ to enable comparisons between varying base values of control cells.

For pressure measurements, a tipless cantilever (MLCT-O10, $f_0 = 10$–20 kHz, $k = 0.03$ N m$^{-1}$, Bruker) was chosen in order to enlarge the contact area of the probed cell. Neutrophils were seeded according to the mentioned protocol, activated with PMA and subsequently incubated for 90 min at 37 °C, 5% CO$_2$ to ensure unmodified cell properties before cell rupture. Afterwards, cells were placed into the aforementioned setup and the cantilever was manually approached to a single cell using 0.2 μm driving steps of the z-piezo motor until cell contact (deflection increase of the cantilever) could be observed. Henceforth, the height of the z-piezo was set constant and the pushing forces of the swelling cell were passively quantified using the Live Tracker function of the JPK SPM Control software (v.5, JPK Instruments). When the pushing forces reached maximal measurable values of the sensor consecutively, the amount of deflection had to be manually reset by readjusting the alignment mirror (reallocation of laser spot on the sensor). This resulted in a vast decline of the deflection data at certain times (dashed lines in Supplementary Fig. 8d), however an effect on the following process could not be observed. Measurements were continued until a rupture of the cellular membrane occurred which was proven afterwards by both a fast breakdown of the deflection data, as well as propidium iodide stainings (Sigma-Aldrich, $c \approx 1$ μM). To calculate the interior pressure of the cell, a phase contrast image of the respective cell was taken directly after reaching the contact point. With this, the contact area $A$ of the cell could be extracted by thresholding the visible cell area and the result was combined with the force $F(t)$ of the deflection measurements to generate the pressure value $p(t) = F(t)/A$.

**Statistics and data analysis.** For data analysis Prism 6 for Mac OS X (v. 6.0 h, GraphPad Software, Inc.), origin (OriginPro8, OriginLab Corporation) and MATLAB (v. R2008a/R2014a; The MathWorks, Inc.) were used.

Statistical analysis was performed with Prism 6 for Mac OS X. For all data sets GAUSS distribution was verified with Shapiro-Wilk normality test if applicable and significance proved by $t$-test or one-/two-way-ANOVA/ Bonferroni's multiple comparisons test (mean ± standard deviation (SD) or standard error of the mean (SEM); *$p < 0.05$; **$p < 0.01$; ***$p < 0.001$; ****$p < 0.0001$). For non-Gaussian distributed data sets Mann-Whitney or Kruskal–Wallis/Dunn's multiple comparisons tests (*$p < 0.05$; **$p < 0.01$; ***$p < 0.001$; ****$p < 0.0001$) were used.

**Time-lapse analysis.** To quantify the change of chromatin area and eccentricity over the time course of NETosis an image segmentation script was developed in Matlab (v. 2014a). Live-cell images generated by wide field fluorescence microscopy were used for the quantification of the chromatin area. This area is a projected chromatin area because it is derived from 2D images. CLSM derived time traces were analyzed similarly and are shown for example in Fig. 1b.

For single cells, areas of high contrast (mainly at the edge between stained regions and the background) were carved out, smoothed and filled to generate an outlined image of each object that could be analyzed. Furthermore, total intensity values were normalized to circumvent problems coming from background fluctuations or different staining intensities. From the resulting area-time curves, four characteristic time points were determined manually as shown in Supplementary Fig. 3a: $t_1$/start of P2: start of first chromatin area increase, $t_D$: end of uniform/isotropic area increase in P2, $t_2$: maximal area in P2, $t_3$/start of P3: second start of increase in chromatin area. Hence, the duration of P1 (activation (0 min) to $t_1$) and P2 ($t_1$ to $t_3$), as well as the diffusion coefficient of the expanding chromatin in P2 ($t_1$ to $t_D$) were calculated. For determination of the diffusion coefficient only cells of population 1 ($t_2 \neq t_3$) were used (see Supplementary Fig. 3a).

Quantitative membrane and chromatin analysis (Fig. 6a–d) was based on the same segmentation procedure and extended: For chromatin swelling analysis (Fig. 6a), detected chromatin areas of each frame were binarized first and stacked to produce the shown density plots. Moreover, each frame was divided in a grid of 15 × 15 pixel sized sub-windows and analyzed by measuring the average additional or lost chromatin area per window resulting in a growth/shrinking vector for each of these segments. The final vector plot results from averaging all frame vectors of a respective segment. For membrane analysis (Fig. 6b–d), the position of the rupture point was manually determined from the first frame after membrane burst and then compared with the computed major axis of the fitted cell ellipsoid. Only cells with a clear single rupture point were analyzed.

**Pressure calculation.** Swelling pressures values shown in Fig. 4d were calculated by using an approach motivated by Mazumder et al.[52] Here, Newtonian properties of the swelling process were assumed and the Stokes-equation was used (effects of inertia were expected to be negligible).

$$\nabla \mathbf{p}(t) = \eta \Delta v(t) \qquad (4)$$

In this case, $p(t)$ describes the inherent pressure of the fluid/chromatin network, $\eta$ is the viscosity of the liquid and $v(t)$ is the velocity field at time point $t$. The physical problem was simplified in two different ways: First, a radial symmetric force field was assumed ($\nabla \mathbf{p} \approx \frac{p}{R}$) and secondly the resistance was assumed to be dominated by viscous forces at the scale of interstices (pores) in the network, which ultimately leads to $\eta \Delta v(t) \approx \frac{\eta}{l_p^2} \frac{dR}{dt}$ with $l_p$ describing the average mesh size and $R(t)$

the radius of the (chromatin) network. Consequently, it follows

$$p(t) \approx \frac{\eta(R(t))}{l_p^2(R(t))} \frac{\mathrm{d}R(t)}{\mathrm{d}t} R(t). \tag{5}$$

The exact value of $\eta(R(t))$ is unknown. However, in previous studies the viscosity of the nucleus of an eukaryotic cell before expansion was determined to be $\eta_0 = 0.1$ Pa s[52]. It is also known that the viscosity of semiflexible polymer network scales $\eta(R) \sim 1/R^3$ with the overall radius $R$ (correlated to the mesh size)[69]. This relation and $\eta_0$ were used to approximate $\eta(R(t))$. Furthermore, to approximate the average mesh size $l_p(R)$, a single DNA strain ($L \approx 2$ m, $d \approx 2.2$ nm) was arranged into a cubic lattice inside a sphere with a radius $R(t)$. $R(t)$ was measured in the time-lapse movies exemplarily shown in Supplementary Movie 2 using $R(t) = \sqrt{A/\pi}$ and $\frac{\mathrm{d}R(t)}{\mathrm{d}t}$ was calculated by fitting the two dimensional diffusion equation $\frac{\mathrm{d}R(t)}{\mathrm{d}t} = \frac{\mathrm{d}(\sqrt{4Dt})}{\mathrm{d}t} = \frac{4D}{2\sqrt{4Dt}} = \sqrt{\frac{D}{t}} = \sqrt{\frac{R^2}{4t^2}} = \frac{R}{2t}$ to the chromatin data between $R(t_1)$ to $R(t_D)$ and $R(t_D)$ to $R(t_2)$ and extracting the resulting slope.

**Code availability**. All Matlab codes used to analyze data are available from the corresponding authors upon request.

## Data availability

Data supporting the findings of this manuscript are available from the corresponding authors upon request.

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

## Acknowledgements

This project was supported by the state of Lower Saxony (life@nano) and the German Research Foundation (DFG grant KR 4242/4-1 and ER 723/2-1). We acknowledge financial support by the open-access funding program of the German Research Foundation and the publication funds of the University Medical Center (UMG). Part of this work was supported by the Cluster of Excellence and DFG Research Center Nanoscale Microscopy and Molecular Physiology of the Brain (CNMPB). We thank Elisa D'Este and Grazvydas Lukinavicius for input on SiR-Hoechst stainings. We thank Andreas Janshoff and Claudia Steinem for fruitful discussions and support. We are grateful for fruitful discussions about active matter with members of the collaborative research center SFB 937 funded by the DFG.

## Author contributions

S.K. and L.E. conceived the study with inputs from M.P.S. E.N. and D.M. performed experiments. E.N., F.R., C.G. and A.E. performed STED microscopy. G.G., A.K.T., J.G. and S.S.S. performed additional staining and inhibitor experiments. E.N., D.M., L.E. and S.K. analyzed data/images. E.N., D.M., L.E., and S.K. wrote the manuscript with inputs from all authors.

## Additional information

**Competing interests:** The authors declare no competing interests.

