## [Peer Review File · Nature Communications]

Reviewers' Comments:

Reviewer #1

Remarks to the Author:

The manuscript addresses the question of the mechanism of NET formation. The manuscript recapitulates on the morphology of NETosis published years ago. It relies on a series of inhibitors, some of them of poor specificity and improperly controlled. Furthermore, a lot of the data are over interpreted and based on indirect experiments.

In more detail:

Figure one present data published in many publications, hence it is confirmatory and can be moved to the supplemental information.

The different stages of "NETosis" were described already ten years ago (J Cell Biol. 2007 Jan 15;176(2):231-4, for example). Also the evaluation of NET formation in single cells is published (J Immunol. 2018 Jan 15;200(2):869-879). This should be clearly stated and, this information should be moved to supplementary informations as well.

The images with STED microscopy also confirm data published using EM and can be moved to the supplementary information.

Chromatin "swelling" has been shown by many groups and does not represent a novel observation. This should be clearly stated in the text-

The authors should define the difference between "swelling" and "expansion" in the context of their manuscript.

The authors use enzymatic inhibitors reproducing data from several groups. The authors should include enzymatic measurements of each of the enzymes that they inhibit and control for diffusion, permeability and half life for each of this inhibitors. The authors should also include controls on the effects of this inhibitors on naïve neutrophils and neutrophils that were incubated with stimulants that activate them, for example to degranulate or phagocytose, but do not induce NETs. These controls are essential for the interpretation of their data since the "timing" might reflect the behavior of the different inhibitors and lead to false conclusions

The analysis of the data (for example Figure 3) is not relevant in absence of appropriate controls.

The manuscript contains many assumptions that should be empirically demonstrated, e.g. the "force" of chromatin on the nuclear membrane.

The authors erroneously identify PMA as a "chemoattractant". PMA has several effects, through MAP kinase activation and is a strong mitogen.

The authors use cytoskeleton inhibitors and demonstrate inhibition of NETs. The NADPH oxidase, essential for NET induction (and confirmed in this manuscript), requires the cytoskeleton for assembly. Hence, controls testing which of this inhibitors affect ROS production have to be included.

The data on "nuclear membrane rupture" relies on indirect evidence and should be interpreted far more carefully than in this manuscript. Or better yer, include direct measurements of nuclear membrane rupture.

The author should also consider alternative explanations to their experiments and design experiments to falsify their hypothesis.

Reviewer #2:

Remarks to the Author:

Neubert et al report on intracellular events occurring during cellular NETosis (the expulsion of DNA during for example an immune response). The authors use a range of microscopy and biochemical tools to indicate important roles of chromatin swelling in this event. The story is nice and well written with convincing data. I would in principle like to see this work published in Nature Communications. There are however still a few issues that should be considered prior to publication.

I sometimes miss quantification of results, such as in figures 2 and 4. Hypothesis are sometimes taken from pure qualitative descriptions of individual cell images, which needs to be supported by solid quantitative data (as done in multiple other occasions such as figures 2b, 4b-d).

I often got confused which measurements were done on fixed and which on live cells, which on CLSM and which on the STED microscope, and how labeling has been done. Please add in main text or in captions.

Most data is performed from CLSM and biochemical experiments. Yet, the abstract sounds like most essential data has been collected with STED microscopy and AFM. The latter data has been crucial, yet the statement in the abstract (... using high resolution fluorescence and atomic force microscopy) is a little bit too one-sided. The authors can be proud of their other data as well.

Page 2, bottom: What is meant by biophysical perspective – not a very explicit expression?

Page 3, bottom: What is meant by fine structures?

Figure 1: What is time 0, addition of drug? Panel b: Hard to see what is shown here – what is inside and outside, where is the cell (could a cell marker be added)?

Page 5, middle: Cannot see filopodia activity!

Figures 2b, 4d,e: Why are the values of the unstimulated cells higher from the very start – do not understand? Why is there chromatin swelling in the unstimulated cells at all?

Supplementary Fig S2c: RICM does not give indication on cell height – therefore I do not understand the statements on page 5, middle.

Page 7, last part: Stage P3 has not been mentioned?

Page 7, bottom: What is “projected area A”?

Page 10, bottom: Did not understand the sentences starting with “Later, these structures ...”.

Reviewer #3:

Remarks to the Author:

The manuscript “Chromatin swelling drives neutrophil extracellular trap release” explores the fascinating phenomenon of NETosis in neutrophils using optical and atomic force microscopy. The manuscript is clear and concise and the conclusions are clearly consistent with the data. However there are some points of concern.

1. The introduction is rather limited – it could be expanded to include more details on what is already known about NETosis as well as its relation to adhesion and migration which are well

studied in neutrophils.

2. It is not clear why t_2 and t_4 are not represented in fig 1d histogram and only in the inset. It is not clear at all how t_0 is set and if this is arbitrary, does the duration of P1 make sense? What is the error associated with determining the times? Were the time demarcations automated (with what criteria)? Similar questions arise for the velocity calculations. In general, a more in-depth discussion of the data analysis should be included. Even if ready-made software routines are used, a description of what they do and how they work is warranted.

3. No information on surface treatment of the substrates are given. Several different substrates are used and it is not clear if any or all of these elicit specific adhesion and if the adhesion is equivalent on all of these.

4. The RICM images are of poor quality, they are not quantified in any way and it not clear that they actually support the conclusion. The cells are very much adhered and it is not clear if they are still able to function correctly. If the point about membrane labelling is crucial, either better RICM images should be provided or another technique should be used to confirm the RICM results. I also recommend providing RICM movies without the overlay of nuclear staining and, for ease of visualization, correcting for the pronounced drift.

5. A major question, not directly dealt with here (which connects to point 1, 3 and 4 above) is the question of how much the extent of adhesion of the cells influence the conclusions. Most techniques presented here depend on strongly adhered cells. As seen in RICM, cells appear to be so tightly adhered that they may not be capable of later migration or deadhesion without leaving behind a substantial part of their membrane. The conclusions will be much stronger if the results are also valid for moderately adhered. In any case, even if no new data can be provided, this point should be discussed.

In conclusion, I recommend publication with revisions.

Response to reviewer comments

We thank all reviewers for their constructive comments and support. Below you can find a point by point response (black) to all comments (blue). Figure numbers changed and therefore we added the term (old) when we refer to the figures from the previous (reviewed) version of the manuscript. The figures shown in this response letter are not labelled with increasing numbers but according to the figure number that we use in the revised manuscript/supplement. In the revised manuscript/supplement we have highlighted changes in yellow.

Reviewer #1 (Remarks to the Author):

1) "The manuscript addresses the question of the mechanism of NET formation. The manuscript recapitulates on the morphology of NETosis published years ago."

The extraordinary morphological changes of cells undergoing NETosis were first published by Zychlinsky et al. in 2004. They provided the incentive and the general basis for the experiments shown in this manuscript. As of today, no dedicated biophysical analysis of this process has been published to our knowledge. In our opinion, the fact that the forces driving this intriguing and dramatic process have remained elusive for such an extended period of time make the questions addressed here more and certainly not less interesting.

2) "It relies on a series of inhibitors, some of them of poor specificity and improperly controlled. Furthermore, a lot of the data are over interpreted and based on indirect experiments."

Some, though by far not all, experimental setups rely on specific and/or general inhibitors of cellular functions. The mixture of very specific inhibitors (such as the MPO-inhibitor ABAH) and inhibitors, which have been very broadly used for decades (see response to point 7 and table1) and which address a multitude of cellular functions (such as sodium azide) is intentional, as it serves to prove our point that the outcome of the interference with the process of NETosis does not rely on the specific inhibitor but rather on the phase this inhibitor is applied in. To further prove this point, we have also chosen a complementary, inhibitor-free approach, which shows the impact of temperature variations on the different phases of NETosis (**figure 3c**). We have emphasized the complementary approach by adding the following sentence (line 207):

"We chose to show the impact of temperature variations on the impact of NETosis as a complementary, inhibitor-free approach to investigate the importance on enzymatic activity."

We agree that each individual inhibitor (in **figure 3b**) alone does not completely rule out other possible conclusions. Therefore, we decided to use multiple inhibitors and the temperature experiments (**figure 3c**) to prove our point that phase 1 is an active biological process, whereas phase 2 is driven by material properties. To further shed more light onto the characteristics of the different used inhibitors we added an extensive set of experiments (**supplementary figure S7**).

Furthermore, in the revised manuscript we have provided additional quantification and evidence for our hypotheses for example in **figure 3a**, **figure 4e**, **supplementary figure S5-S9** (see specific responses below).

Finally, we have added new data giving direct evidence of the reduction of intracellular ATP levels during P1 and a plateau of ATP levels during P2, which further adds to our hypothesis that P1 is an energy-dependent process, while P2 is not (**figure 3a**).

3) "Figure one present data published in many publications, hence it is confirmatory and can be moved to the supplemental information. The different stages of "NETosis" were described already ten years ago (J Cell Biol. 2007 Jan 15;176(2):231-4, for example)."

Figure 1 presents the basis of our classification of the different stages of NETosis and is therefore essential to the understanding of our publication. Therefore, moving this figure to the supplement would cause great difficulties in understanding the rest of the manuscript, which is why we have chosen to leave figure 1 in the main part of the document.

The publication reviewer 1 refers to of Fuchs et al. (a very important publication which we cite at several points in our manuscript), shows pictures of single exemplary cells undergoing NETosis at different time-points.

However, they skip from $t = 4$ min to $t = 79$ min, thus omitting “Phase 1” completely, which to our current knowledge is crucial to NET production. This publication does not provide a continuous time-course of chromatin morphology during NETosis. Neither this publication nor several others, which phenomenologically describe chromatin decondensation [1-3] have provided a systematic and quantitative evaluation of these chromatin changes over time and used this to characterize different phases of NETosis. Nor have these publications provided any link of their morphological observations to mechanistic processes.

Therefore, we think that our approach at characterizing chromatin morphology of a large number of cells, providing a classification of the different phases and developing a biophysical model based on this data is not only novel but highly valuable. It could change the way scientists think about NETosis and other cellular processes and how they design their experiments.

4) “Also the evaluation of NET formation in single cells is published (J Immunol. 2018 Jan 15;200(2):869-879). This should be clearly stated and, this information should be moved to supplementary informations as well.”

As explained above, our analysis goes far beyond a merely descriptive “evaluation of NET formation in single cells” in so far as it provides – to our knowledge for the first time – a link between chromatin dynamics (analysed and quantified in a large number of single cells) and different functional phases of NETosis.

However, to put more emphasis on the fact that NETosis has been observed in single cells, we have added the following sentence, including the reference mentioned by the reviewer (line 66):

“Chromatin decondensation had been described qualitatively since the discovery of NETs[1, 4, 5] and NET formation has been evaluated both in high-throughput approaches as well as on the single-cell level[1-3]”

5) “The images with STED microscopy also confirm data published using EM and can be moved to the supplementary information.”

To our knowledge we here show the first fluorescence super-resolution images of NETs and NETotic cells. It is difficult to compare EM images to STED microscopy as in EM cells are fixed, dehydrated and analysed under ultra-high vacuum conditions. While EM has a superior resolution, due to the nature of this harsh treatment, EM images are highly susceptible to artefacts, especially when trying to visualize dynamic processes. While the STED images provided in **figure 1b** (old) showed fixed cells, **figure 4c** provides live-cell STED images which constitute a completely novel method to image NETosis in a much less artefact-prone way.

Therefore, we feel that the images from **figure 4c** should be kept in the main part of the document because it shows the homogenous chromatin network in the cell.

We agree that **figure 1b** (old), even though the first STED image of a NET, does not add a lot of information to the mechanism of NET release. Hence, we moved it to the supplement, as requested (see **supplementary figure S2**).

6) “Chromatin “swelling” has been shown by many groups and does not represent a novel observation. This should be clearly stated in the text-The authors should define the difference between “swelling” and “expansion” in the context of their manuscript.”

We agree with reviewer 1 that chromatin decondensation has been observed by most groups dealing with NETosis. Indeed, we clearly state that chromatin decondensation is a hallmark that has been described since the discovery of NETosis (see line 66).

However, “swelling” in the context of this paper refers to a defined physicochemical process - not to the phenomenological observation of chromatin decondensation. It is therefore a novel concept, which has not been addressed in previous publications. In our point of view our paper presents a paradigm shift for this field: Chromatin expansion is not just the consequence of biochemical processes. It is an active player in morphological changes and NET release and dominates phase 2 and 3.

To facilitate a distinction between the descriptive term of chromatin “expansion” or “decondensation” and the term “swelling” we have clarified this in the manuscript. We first use the term expansion/decondensation for the morphological description of this process. After introducing the concept and the evidence of swelling as the major driving force of NETosis we consequently use the term “swelling” throughout the rest of the manuscript.

7) “The authors use enzymatic inhibitors reproducing data from several groups. The authors should include enzymatic measurements of each of the enzymes that they inhibit and control for diffusion, permeability and half life for each of this inhibitors.”

It is correct that the inhibitors used in our control experiments have been used by many groups and are well established in the field of NET research, which was our main reason to conduct our control experiments with them. For all of these inhibitors, the pharmacokinetic and pharmacodynamic properties are largely known and published. We have summarized some of the literature (see **table 1** below) to show that these molecules are suitable controls/inhibitors with known pharmacological profiles.

Table 1: Inhibitors and their use in the literature

Inhibitor (used concentration)	General information	Inhibitory mechanism	Biochemical data	Evidence in neutrophils/ NETosis
4-Aminobenzoic Acid hydrazide (4-ABAH) (100 μ M)	 Molecular weight: 151.2 g/mol	Irreversible inhibition of myeloperoxidase (MPO) at high H_2O_2 concentrations/ low oxygen concentration [6] Extracellular production of HClO/ low H_2O_2 concentrations \rightarrow e.g. competing with chloride and acting as a competitive substrate [6]	Half life Blood half-life estimated to be approximately 10 min based on chemical structure/ small molecule [7] IC₅₀ values Purified MPO/ peroxidation: 2.2 μM [8] Purified MPO/ chlorination activity: 0.3 μM [8] Neutrophils: 16 μM (without PMA), 2.2 μM (with PMA) [8]	Neutrophils Effect on other enzymes involved in H_2O_2 metabolism \rightarrow no inhibition of catalase or glutathione peroxidase, no effect on O_2^-, no effect on degranulation (release of lysozyme and β-glucuronidase) [8] NETosis Slows down NETosis: significant decrease and delay of PMA-induced NET formation (at 500 μM) [9] Time-dependent decrease of MPO activity of azurophilic (AZ) granules (ABAH (500 μM) + H_2O_2) [10]
2-deoxyglucose (2-DG) (5 mM)	 Molecular weight: 164.16 g/mol	Competitive inhibition of phosphoglucoisomerase (rat kidney) [11] Noncompetitive inhibition (2DG-6-PO₄) of hexokinase (bovine heart hexokinase) [12] \rightarrow depletion of ATP 2-DG retards uptake of glucose [13]	Half life Half-life of 2DG in plasma is 48 min [17] IC₅₀ values 1-7 mM (different colorectal cancer cell lines) [18] Permeability/ Uptake Uptake by glucose transporters and direct phosphorylation to	Inhibits DNA release induced by PMA [21, 22] or amyloid fibrils [21] No effect on viability (PMA + 2DG) [21]

		Influence on glucose metabolism - Pentose phosphate pathway (6-glucose dehydrogenase) [14] - Glycogen production [15] - Glycosylation [16]	2DG-6-PO ₄ by hexokinase [19] Uptake of ¹⁴ C-2DG or 2DG-IR within 2-4 min [20]	
Sodium azide/ NaN₃ (3 mM)	Na^+ $\text{N}^+ \equiv \text{N}^- \equiv \text{N}^-$ Molecular weight: 65.01 g/mol	Inhibition of mitochondrial respiration Competitive Inhibition of ATP hydrolysis (interaction: β -phosphate of ADP/residues of ADP-binding catalytic subunit β_{DP} , bovine F1-ATPase)[23] Inhibition of respiratory chain complex IV/ cytochrome c oxidase (binds to the metal binding sites (heme a ₃ and Cu _B)), from bovine heart [24] Other metalloproteins - catalase[25] - SOD[26] - MPO[27]	IC₅₀ values 10 μ M (AI3) IC₂₀ values 50 μ M (E. coli)[28] Permeability/uptake Maximal activity time 10 min (grasshopper embryo cells)[29]	Neutrophils Enhanced migration (1-10 μ M) due to nitric oxide formation[30] NETosis Primarily inhibits azurophilic granule proteins (MPO) → inhibition of NETosis via ROS [31]

Again, we would like to emphasize that we purposefully used a mixture of very specific inhibitors (such as the MPO-inhibitor 4-ABAH) and inhibitors of general cellular functions (such as sodium azide), as this clearly shows that the outcome of NET inhibition does not depend on the specificity of the inhibitor but on the phase of NETosis it is applied in.

It is out of the scope of this project and in our opinion also not adding new information (see **table 1** and arguments below) to establish experimental setups to determine diffusion, permeability and half-life for all used chemical components. However, it is our understanding that reviewer 1 is concerned about inhibitor characteristics because they might influence the outcome of our experiments, especially the time-resolved data, which is why we here provide detailed new data to refute any uncertainties regarding our inhibitors (see below).

First of all, it should also be considered that unexpected or prolonged uptake kinetics etc. (e.g. in **figure 3b**) would cause a bias to our disadvantage, as they would have blurred the boundaries between Phase 1 and 2. And yet, the different inhibitors show similar outcomes (in **figure 3b**) indicating that uptake kinetics etc. are not significant confounders in this experiment. Nevertheless, we have devised a large comprehensive series of experiments to show that each of the used enzymatic inhibitor does indeed exert its action in the expected way and without a significant time-delay (see new **supplementary figures S7**):

- 1) Inhibition of ROS production by sodium azide: We used a luminescence-based luminol assay to show that sodium azide as a general inhibitor of both the respiratory chain and other metalloproteins inhibits ROS production in neutrophils in a fast and durable fashion.

Figure S7a, Effect of 3 mM sodium azide (NaN_3) on PMA-induced ROS production of human neutrophils determined by chemiluminescence of luminol. NaN_3 inhibits ROS directly after addition and enables a stable inhibition for at least 30 min, as shown in **supplementary figure S9c**. Experimental setup comparable with the setup used for the experiments shown in **figure 3b**. PMA = 100 nM. $N = 1$ (triplicates). Mean \pm SD.

- 2) Inhibition of MPO activity by 4-ABAH: We show that the MPO-inhibitor 4-ABAH inhibited purified MPO within 1 min and stable within 15 min after PMA activation:

Supplementary figure S7: d, inhibitory effect of 100 μM 4-aminobenzoic acid hydrazide (4-ABAH) on MPO activity. **Left**, 4-ABAH inhibits purified MPO significantly and stable after 1 min for at least 30 min. $N = 3$. Mean \pm SD. **Right**, 4-ABAH inhibits PMA (100 nM)-induced MPO activity significantly and stable after 15 min and 30 min cell incubation followed by complete cell lysis for MPO activity measurements. Experimental setup comparable with the setup used for the experiments shown in **figure 3b**. Statistics: Two-way ANOVA (Bonferroni's multiple comparisons test, $**p < 0.01$, $***p < 0.001$). $N = 3$. Mean \pm SEM.

- 3) ATP-reduction after addition of 2-Deox-Gluc: We chose here to measure ATP levels after the addition of 5 mM 2-Deox-Gluc as this assay provides a sensitive assay for the effect of 2-Deox-Gluc, as compared to the also frequently used lactate-secretion assay which is much less sensitive[32]. We could show that

there was a significant reduction of 20 % in ATP- levels 60 min after application of 2-Deox-Gluc in non-stimulated cells (**supplementary figure 7b**).

Supplementary figure S7b and c: **b**, ATP levels of unstimulated neutrophils after incubation with 5 mM 2-Deoxy-glucose (2-Deox-Gluc) for different time periods. 2-Deox-Gluc reduces ATP levels already after short incubation and significantly after more than 60 min compared to untreated cells ($t = 0$ min). Statistics: One-way ANOVA (Bonferroni's multiple comparison test, $**p < 0.01$, $***p < 0.001$, $****p < 0.0001$). $N = 3$. Mean \pm SEM. **c**, ATP levels of PMA (100 nM)-stimulated neutrophils with and without incubation with 5 mM 2-Deoxy-Gluc for different time periods. 2-Deoxy-Gluc significantly decreases the ATP levels after more than 15 min PMA stimulation compared to exclusive PMA treatment ($t = 0$ min). PMA stimulation alone decreases ATP levels by more than 70 %. Experimental setup comparable with the setup used for the experiments shown in **figure 3b**. Statistics: Two-way ANOVA (Bonferroni's multiple comparisons test, $*p < 0.05$, $**p < 0.01$, $***p < 0.001$, $****p < 0.0001$). $N = 3$. Mean \pm SEM.

The effect of 2-Deox-Gluc on non-stimulated cells is important, however for our experiments the energy levels of stimulated cells are crucial. As shown in new **supplementary figure 7c**, activation by PMA itself already reduced the cellular ATP levels by more than 50 % within 15 min. Adding 2-Deox-Gluc further reduced it by additional 50 % ($t=60$ min). After around 60 min a stable plateau was reached. We added a simplified version of this figure to the revised main manuscript (new **figure 3a**).

Figure 3a: ATP levels in stimulated neutrophils decrease during P1 and reach a plateau in P2. Inhibition of glycolysis further reduces the ATP levels (**supplementary figure S7c**). $N = 3$. Mean \pm SEM.

This data set clearly shows that 2-Deox-Gluc reduces ATP levels, as requested by the reviewer. The concentration of 2-Deox-Gluc (5 mM) in relation to glucose (11 mM) and the mechanism (competitive inhibition) explains why one should not expect 100 % reduction. In our experimental setup this level of ATP reduction was sufficient to inhibit NETosis in phase 1. Additionally, we gained the new insight that activation of cells by PMA further reduces

ATP levels and when combined with 2-Deox-Gluc this effect is additive. In summary, this extended data set further proves our hypothesis that a reduction of energy supply in phase 1 inhibits NETosis and that phase 2 is not (or less) energy-dependent.

We have modified the text on page 7-8 to include all these different new experiments.

8) “The authors should also include controls on the effects of this inhibitors on naïve neutrophils”

We have added our data on the influence of the inhibitors on naïve neutrophils (no effect as expected, see new column in **figure 3b** and **figure 5b/e**) and modified the text accordingly. Additionally, we also performed an assay to control for cytotoxicity of all used enzymatic inhibitors as well as the inhibitors of the cytoskeleton (**supplementary figure S7e**). As expected for all used inhibitors values <10 % were found similar to values for the (control) solvent (DMSO, 1% maximal concentration used to solve inhibitors).

Figure 3b: Metabolic inhibitors (sodium azide/ 3 mM, 2-deoxy-D-glucose/ 5 mM, 4-aminobenzoic acid hydrazide/ 100 μM) influence NET formation determined as relative number of decondensed nuclei after 180 min compared to activation with PMA only. None of the inhibitors induces NETosis of unstimulated cells. N = 3 donors. Statistics: two-way ANOVA (Bonferroni’s multiple comparisons test; * p < 0.05; ** p < 0.01; **** p < 0.0001; ns = not significant). Mean ± SEM.

Supplementary figure 7e: Toxicity of used inhibitors on human neutrophils detected by release of lactate dehydrogenase (LDH) relative to complete cell lysis. All inhibitors were tested for the concentrations used in this study and the maximal incubation time of 3h. All inhibitors show less than 10% cell toxicity, which is below the toxicity of 1% DMSO with 10.2% (maximal solvent concentration in inhibitor studies). N = 2. Mean ± SD.

We have modified the text on page 8 to include these results and refer to them in the supplement.

9) “and neutrophils that were incubated with stimulants that activate them, for example to degranulate or phagocytose, but do not induce NETs. These controls are essential for the interpretation of their data since the “timing” might reflect the behavior of the different inhibitors and lead to false conclusions”

We have performed a phagocytosis assay to show the influence of our enzymatic inhibitors on other cellular functions apart from NET formation. As expected from the literature, 2-Deox-Gluc inhibits phagocytosis/particle uptake, as this cellular process is also energy-dependent[32]. Addition of 4-ABAH (inhibition of MPO) to the phagocytosis assay also shows a decrease, while sodium azide does not influence particle uptake in our experiments.

Supplementary figure 7f: Effect of NaN₃ (3 mM), 4-ABAH (100 μM) and 2-Deox-Gluc (5 mM) on uptake of FITC-labeled *E.coli* BioParticles. The particle uptake is not affected by NaN₃, but clearly decreased in presence of 4-ABAH and 2-Deox-Gluc. Calculation based on confocal imaging of fixed samples after incubation with BioParticles for 30 min. Staining: red = F-actin/ Phalloidin555, blue = chromatin/Hoechst, green = FITC-labeled *E.coli* BioParticles. Scale = 10 μm. N = 2 (n = 60 cells/condition/N). Mean ± SD.

We have added the result of this assay to the supplement (see **supplementary figure S7f**).

10) “The analysis of the data (for example Figure 3) is not relevant in absence of appropriate controls”.

We are not sure how to interpret this rather general sentence. **Figure 3** (old) contains different data sets that in our view contain appropriate controls wherever possible. In **figure 3a** (old) different inhibitors were used (as controls) to test our hypothesis of an active phase 1 and a passive phase 2. In the new **figure 3b** we have added another control (no stimulation + inhibitor). **Figure 3b-f** (old) contained temperature experiments, quantification of chromatin area, kinetic analysis of the rupture time point and a theoretical quantification of the pressure. **Figure 3e** (old) shows the first STED image of the chromatin in a living neutrophil just before the membrane ruptured (late phase 2). For these figures, we do not know if and what kind of additional control the reviewer would suggest. In our opinion, in every experiment for which a reasonable control was possible we performed the control experiments.

11) “The manuscript contains many assumptions that should be empirically demonstrated, e.g. the “force” of chromatin on the nuclear membrane.”

There is, to our knowledge, currently no method to directly measure intracellular forces at the nuclear envelope non-invasively, as the isolation of nuclei represents a very strong manipulation of the neutrophils. In the paper (**figure 2-5**) we use different biochemical control experiments and imaging techniques to rule out that other processes dominate this process except swelling (as the major driving force) beginning with phase 2. The temporal correlation between chromatin expansion (**figure 2d**) and a) nuclear envelope defects (**figure 2c**) and b) the onset of inhibition failure (**figure 3b** and **figure 5b/c**) and c) softening of the cell (**figure 5d**) and softening of the membrane (**figure 5e**) and the size of chromatin area directly after nuclear envelope rupture (**figure 2c**, **supplementary figure S6**) support the swelling hypothesis. Another strong evidence is the isotropic expansion of the chromatin (**figure 2d**) that also takes place if the energy supply is shut down. This conclusion does not rule out other biochemical influences but our paper identifies swelling as a major driving force after the biochemical processes in phase 1.

Additionally, we collected further evidence that there are forces involved by performing a new type of AFM experiment (**figure 4e** and **supplementary figure S8d**). In short, a tip-less cantilever was positioned above a cell performing NETosis. This procedure has been used by other groups in a different biological context to measure

rounding forces during mitosis (Stewart et al. Nature 2011). As shown in **figure 4e** the cell exerts a force on the cantilever while they round up and deflects the cantilever until the cell finally ruptures. This force could be either due to the swelling pressure of the chromatin or active rounding of the cell. We have shown in the temperature experiments (**figure 3c**), the energy supply experiments (**figure 3a/b**) and the cytoskeleton interference experiments (**figure 5b/c**) that phase 2 does not primarily depend on biochemical factors. Consequently, the forces we measured can be attributed to the swelling process.

Figure 4e: Neutrophils undergoing NETosis exert an increasing pressure on a fixed AFM cantilever until they rupture (end point of measurement). $N = 5$ ($n = 14$ cells).

12) *“The authors erroneously identify PMA as a “chemoattractant”. PMA has several effects, through MAP kinase activation and is a strong mitogen.”*

We thank the reviewer for this comment and have specified the action of PMA on neutrophilic granulocytes to give credit to the pleiotropic effects of PMA (line 298-300).

13) *“The authors use cytoskeleton inhibitors and demonstrate inhibition of NETs. The NADPH oxidase, essential for NET induction (and confirmed in this manuscript), requires the cytoskeleton for assembly. Hence, controls testing which of this inhibitors affect ROS production have to be included.”*

The reviewer raises a very important and interesting point. It has been shown for several cytoskeleton inhibitors that they have an effect on the NADPH oxidase. Therefore, it is possible that the cytoskeletal inhibitors we here use may have off-target effects on the NADPH oxidase and thus cause a bias to our results.

We have, therefore, followed the suggestion of reviewer 1 and tested how our cytoskeletal inhibitors influence ROS production. Here, we show that, in line with the literature, jasplakinolide indeed reduces ROS formation [33]. Y-27623 had no measurable influence on the ROS levels (**supplementary figure S9c**). Additionally, it is known that latrunculin, an inhibitor of actin-polymerisation, increases ROS production by neutrophils, which could also be corroborated by us (**supplementary figure S9c**) [33]. For cytochalasin D (another inhibitor of actin polymerization) we observed a slight increase of ROS.

Furthermore, for jasplakinolide and Y-27623, we saw a significant reduction in NET-formation (**figure 5c**). Additionally, we investigated a new cytoskeletal inhibitor that interfere with actin polymerization (Latrunculin A) and have moved the supplementary data on Cytochalasin D to the main part of the manuscript (new **figure 5b**). In early phase 1 NETosis was inhibited. In phase 2 there was no significant impact on NETosis. These results (new **figure 5b**) further support that the cytoskeleton does not play a role in the final release of NETs.

Thus, while the effect on ROS production was fundamentally different for all four used inhibitors, all four showed a similar impact on NETosis (**figure 5b/c**) from which it can be concluded that although cytoskeletal inhibitors may have complex off-target effects on the NADPH oxidase, the direct effect on the cytoskeleton outweighs these effects with respect to the effect on NETosis.

We have added this information to the manuscript on page 11-13.

Supplementary figure 9c: Influence of actin cytoskeletal inhibition on PMA (100 nM)-induced ROS production of human neutrophils determined by the chemiluminescence of luminol. Latrunculin A (dark red) increases ROS, while Cytochalasin D (violet) and Y-27632 (red) have no or only slight effects on ROS production in the concentrations used for the experiments shown in **figure 5b/c**. In contrast, Jasplakinolide (yellow) shows a strong inhibitory effect. As controls, the ROS levels of unstimulated cells (blue), cells after addition of 1% DMSO (dark green, used for Jasplakinolide experiments) and NaN_3 (green) are shown. $N = 3$. Mean \pm SEM.

Figure 5b: Inhibition of NET formation with the F-Actin polymerization inhibitors Cytochalasin D (100 nM) and Latrunculin A (1 μM), significantly reduces the formation of NETs (measured as % relative number of decondensed nuclei after 180 min compared to activation with PMA only) in P1, while P2 is not affected. Statistics: two-way ANOVA (Bonferroni's multiple comparisons test; * $p < 0.05$; **** $p < 0.0001$; ns = not significant). $N = 3$ -5 donors. Mean \pm SEM.

14) "The data on "nuclear membrane rupture" relies on indirect evidence and should be interpreted far more carefully than in this manuscript. Or better, include direct measurements of nuclear membrane rupture."

We agree that direct measurements of membrane rupture would be desirable. However, we labelled the envelope that contains the (labelled) chromatin and used defects in the envelope in addition to (early) asymmetric expansion of chromatin through the defect as an indicator of rupture. We added a quantification of such images in different phases of NETosis. Unfortunately, we are not aware of a more direct method. Similar approaches have been used by other groups to quantify nuclear envelope rupture (e.g. “Automated analysis of cell migration and nuclear envelope rupture in confined environments, Elacqua, Lammerding et al. PLOS 2018). It has also been described in the literature (e.g. Fuchs et al. JCB 2007, cited by us) before that the nuclear envelope loses its integrity during NETosis.

In **figure 2c** we show with fluorescence microscopy (lamin and chromatin staining) that in phase 1 the nuclear envelope is intact but opens at one or more positions when phase 2 begins. We added a quantification in **supplementary figure S6** (see below, reviewer 2). Furthermore, the chromatin area of cells in which the nuclear envelope just ruptured fits very well with the onset of phase 2 (see **figure 1 and supplementary figure S6**).

15) “The author should also consider alternative explanations to their experiments and design experiments to falsify their hypothesis.”

After we introduce the phase classification in **figure 1** the rest of the paper (**figure 2-6**) is used to verify our hypothesis and falsify other hypothesis (for instance: active transport, energy dependent phase 2, cytoskeleton dominated release,...). Therefore, we are convinced that we tested all thinkable alternative explanations and ruled them out.

Reviewer #2 (Remarks to the Author):

1) “Neubert et al report on intracellular events occurring during cellular NETosis (the expulsion of DNA during for example an immune response). The authors use a range of microscopy and biochemical tools to indicate important roles of chromatin swelling in this event. The story is nice and well written with convincing data. I would in principle like to see this work published in Nature Communications. There are however still a few issues that should be considered prior to publication.”

We thank reviewer 2 for his favourable opinion of our manuscript.

2) “I sometimes miss quantification of results, such as in figures 2 and 4. Hypothesis are sometimes taken from pure qualitative descriptions of individual cell images, which needs to be supported by solid quantitative data (as done in multiple other occasions such as figures 2b, 4b-d).”

In general, our work is based on single-cell analysis. Typically, we show an exemplary single cell data set and then we quantify different aspects for multiple cells from multiple donors. In those cases where this had not been done, we have now added a quantification for the two sub-figures mentioned by the reviewer (new **supplementary figure S6 and S9a**, see below).

Supplementary figure S6: Overlay of chromatin (blue) and lamin B1 (green) of unstimulated and NETotic human neutrophil (CLSM, fixed samples). Average chromatin area at rupture point of the surrounding lamin B1 (t_1) is $41.9 \pm 6.3 \mu\text{m}^2$ ($n = 26$ cells from two donors) and in good agreement with the chromatin area at t_1 determined during live cell imaging (figure 1a/b). At this time point, the nuclear envelope rupture events increase (t_1 / start P2) until a high rate of rupture events (small whole up to full loss of the Lamin B1 surrounding in P2 and P3. Calculation based on CLSM images of fixed samples. $N = 2$ donors (100 cells/ condition/ N). Mean \pm SD. Scale = $10 \mu\text{m}$.

Supplementary figure S9a: Quantification of F-Actin disassembly. The mean fluorescence intensity of F-actin decreases with time during NETosis. At the same time the heterogeneity decreases, which is a measure for F-actin structure and not biased by bleaching. Thus, F-actin gets disassembled during NETosis. $N = 1$ donor.

3) "I often got confused which measurements were done on fixed and which on live cells, which on CLSM and which on the STED microscope, and how labeling has been done. Please add in main text or in captions."

As requested we have specified the method of visualization (CLSM, STED, conventional microscopy) and whether cells were fixed or alive in all figure captions. We thank the reviewer for pointing this out because that information was indeed missing in the captions.

4) "Most data is performed from CLSM and biochemical experiments. Yet, the abstract sounds like most essential data has been collected with STED microscopy and AFM. The latter data has been crucial, yet the statement in the abstract (... using high resolution fluorescence and atomic force microscopy) is a little bit too one-sided. The authors can be proud of their other data as well."

We agree that our CLSM data is very important. STED microscopy was not performed before in the context of NETosis and adds novel information about the mesh size and heterogeneity of the expanding chromatin. We have

modified our abstract accordingly and removed the term high resolution because it is included in the term fluorescence microscopy.

5) “Page 2, bottom: What is meant by biophysical perspective – not a very explicit expression?”

We modified the introduction to make this expression more understandable (page 2). The term “biophysical” is of course very general and we used it in different circumstances to distinguish the existing biochemical perspective in the NETosis field from the novel perspective presented in our manuscript.

6) “Page 3, bottom: What is meant by fine structures?”

We thank the reviewer for this comment and modified the text to make it clearer. In the revised version, the term “architecture” is used to emphasize that the spatial organization of the chromatin is meant.

7) “Figure 1: What is time 0, addition of drug? Panel b: Hard to see what is shown here – what is inside and outside, where is the cell (could a cell marker be added)?”

Time 0 min is the addition of the drug (e.g. PMA). We have added arrows to clarify that (see also response 3, reviewer 3). The experiments are typically performed in (heated) chambers and it takes some time to assemble everything after adding the drug, putting it on the microscope and finding the right focus. Therefore, the movies start a few minutes later but during this time period nothing relevant happens (no change of chromatin morphology) as verified by experiments in open chambers.

Figure 1b (old) shows the chromatin after a cell performed NETosis (i.e. a NET). To our knowledge, this is the first super resolution image of a released NET. It shows all chromatin (intracellular and extracellular) of a cell that performed NETosis but from this image alone one cannot conclude unambiguously what is inside or outside. Reviewer 1 has requested that **figure 1b** be moved to the supplement (new **supplementary figure S2**) and we have complied with his wish because our main focus is the processes before the chromatin gets out of the cell. The important STED microscopy image of the completely chromatin-filled cell remains in the manuscript, of course (**figure 4c**). We agree that it would be nice to combine the DNA staining with a marker for the cell membrane, however establishing two-color STED would be challenging and would not add much additional information.

8) “Page 5, middle: Cannot see filopodia activity!”

Filopodia activity is indeed not well visible in **figure 2a**. We have therefore removed the reference from **figure 2a** in the context of filopodia activity. However, the new **supplementary figure S4b** shows filopodia. For ease of identifying the filopodia we have added arrows to mark the filopodia.

Supplementary figure S4b: Characteristic behavior of the cell membrane during NET formation observed by time-lapse CLSM (blue = DNA, red = membrane). Cells form membrane extensions (arrows) in late PI. Scale = 10 μm .

9) “Figures 2b, 4d,e: Why are the values of the unstimulated cells higher from the very start – do not understand? Why is there chromatin swelling in the unstimulated cells at all?”

Neutrophils are typically round. When stimulated they adhere and flatten. NETotic cells are therefore first flatter than their (unstimulated) control cells but in the end attain a spherical shape again (similar to the non-stimulated control cells) before they rupture. We have modified the figure caption accordingly and added a schematic to clarify this point.

Figure 2b, Cell height as measured by atomic force microscopy (AFM) on live neutrophils. PMA stimulated cells adhere and flatten (compared to the control cells that stay more or less round) and then round up ($> 8 \mu\text{m}$) again in phase 2. $N = 3$. Mean \pm SEM

10) “Supplementary Fig S2c: RICM does not give indication on cell height – therefore I do not understand the statements on page 5, middle.”

Indeed, RICM provides information about the interface between cell and substrate and this sentence was in the wrong context. We have clarified that on page 15-16 and added a few more sentences about the RICM results (see additional data and question by reviewer 3).

11) “Page 7, last part: Stage P3 has not been mentioned?”

The beginning of P3 represents the burst of the cell (see **figure 1b** for the definition) and, consequently, the end NETosis. It is therefore not possible to determine an end-point of P3 and, in consequence, the duration of P3 cannot be investigated in the context of temperature-dependency. We have clarified that in the manuscript.

12) “Page 7, bottom: What is “projected area A”?”

This term refers to the fact that chromatin area is calculated from a 2D image. However, we feel that this term is rather confusing as noticed by the reviewer and have therefore chosen to omit it, as the description of the method gives information about how the area was calculated. We have inserted a sentence to clarify this in the materials and methods part.

13) “Page 10, bottom: Did not understand the sentences starting with “Later, these structures ...”.”

We thank the reviewer for this hint and rewrote this part (page 11) to clarify that microtubule structures disappear with the begin of phase 2.

Reviewer #3 (Remarks to the Author):

1) “The manuscript “Chromatin swelling drives neutrophil extracellular trap release” explores the fascinating phenomenon of NETosis in neutrophils using optical and atomic force microscopy. The manuscript is clear and concise and the conclusions are clearly consistent with the data. However there are some points of concern.”

We thank the reviewer for positive view of the paper and his valuable suggestions!

2) “The introduction is rather limited – it could be expanded to include more details on what is already known about NETosis as well as its relation to adhesion and migration which are well studied in neutrophils.”

We have expanded the introduction (page 2) to include more details about what is already known about NETosis, especially with respect to the role of adhesion. In fact, while there seems to be evidence for a role of integrins, especially Mac-1, in NETosis, many questions still remain to be answered in this context. In fact, this question is a subject of ongoing studies in our lab.

3) "It is not clear why t_2 and t_4 are not represented in fig 1d histogram and only in the inset. It is not clear at all how t_0 is set and if this is arbitrary, does the duration of PI make sense?"

What is the error associated with determining the times? Were the time demarcations automated (with what criteria)? Similar questions arise for the velocity calculations. In general, a more in-depth discussion of the data analysis should be included. Even if ready-made software routines are used, a description of what they do and how they work is warranted.

At $t = 0$ min the stimulant was added and at t_0 the movie was started. After adding the stimulant it typically takes a few minutes to put the (closed) heating chamber on the microscope and find the focus. During this period the chromatin area stays constant. t_0 did not add to the understanding of the manuscript because as well as t_4 (maximum after NET release). We thank the reviewer for pointing out that it might only confuse the reader. Therefore, we decided to completely remove these time points from our experiments and changed **figure 1b**. There is no significant error in determining the experimental time (1 s) with respect to the time scale of the biological process (minutes-hours). However, the frame rate (1 image/minute) determines the time resolution. Again, this error is expected to be far below the length of phase 1 (around an hour) and phase 2 (around an hour). The start of chromatin swelling and the cell rupture is characterized by a steep jump in chromatin area and was manually identified (see also **supplementary figure 1**). We expect the error to be on the order of the time resolution (1 min).

We have revised the data analysis part in the materials & methods section. Specifically, we defined time point 1 as requested by the reviewer.

4) "No information on surface treatment of the substrates are given. Several different substrates are used and it is not clear if any or all of these elicit specific adhesion and if the adhesion is equivalent on all of these."

The materials section contains information about the surfaces we used. In most experiments (if not stated otherwise) standard plastic substrates (surface activated, ibidi treat) were used. The reviewer raises the interesting question of adhesion and we addressed this very important aspect in detail below (point 6 below). In short, our results show that the surface does not have an important impact on NETosis in our experimental setup.

5) "The RICM images are of poor quality, they are not quantified in any way and it not clear that they actually support the conclusion. The cells are very much adhered and it is not clear if they are still able to function correctly. If the point about membrane labelling is crucial, either better RICM images should be provided or another technique should be used to confirm the RICM results. I also recommend providing RICM movies without the overlay of nuclear staining and, for ease of visualization, correcting for the pronounced drift."

We have generated new RICM images of better quality (**supplementary movie M9**). Furthermore, we have now used the new RICM images from **supplementary figure S4c** to visualize the extent of adhesion of the neutrophils on glass (see question below).

Supplementary figure 4c and d: **c**, Representative images of human neutrophils undergoing PMA-induced NET formation recorded with real-time reflection interference contrast microscopy (RICM) on glass. Images allow the label-free analysis of the cell/surface contact area (black = cell closer to the surface, white = further away from the surface). During NETosis, the cells round up, leave membrane closely bound to the substrate behind (arrows, $t = 90$ min) and expel the NET (see also **movie M9**). Scale = 10 μm . **d**, Overlay of real-time RICM with immunofluorescence (blue = chromatin) to verify NET release ($t = 70/90$ min) during time-lapse RICM imaging. Scale = 10 μm .

We also included a novel video on the ibidi treat slides, which was used for all live-cell experiments, to provide better comparability of the experiments (**supplementary movie M10**). As requested, we have also removed the overlay with the nuclear staining (**supplementary movie M9** (old)). In this case we used normal glass slides because they enabled the best optical (RICM) quality. We chose, to keep the overlay images in **supplementary figure S4d** to show that the cells were actually undergoing NETosis.

We agree that in the previous version of the manuscript the usefulness of RICM was not completely clear. The new data presented here makes it much clearer what can be learned from RICM: 1) One can clearly see (**movie M9, M10 and supplementary figure S4d/c**) in the RICM movies that the cell leaves membrane behind and rounds up, which validates the fluorescence microscopy data. 2) RICM allowed us to determine the adherence of neutrophils on different substrates (new **supplementary figure S5**, see response to question 5 below). We have rewritten this part in the manuscript (page 15-16) and addressed this point.

6) “A major question, not directly dealt with here (which connects to point 1, 3 and 4 above) is the question of how much the extent of adhesion of the cells influence the conclusions. Most techniques presented here depend on strongly adhered cells. As seen in RICM, cells appear to be so tightly adhered that they may not be capable of later migration or deadhesion without leaving behind a substantial part of their membrane. The conclusions will be much stronger if the results are also valid for moderately adhered. In any case, even if no new data can be provided, this point should be discussed.”

The reviewer raises an extremely interesting question. We addressed this question by performing NETosis experiments on surfaces with tunable adhesion/passivation.

Our data show that adhesion is not a crucial factor for the outcome of NETosis (new **supplementary figure S5**). It does not rule out that adhesion plays a role in the threshold at which the NETosis pathway is pursued *in vivo* but in our experimental setup the role of adhesion is smaller than we expected.

We tested different surfaces that were passivated with Poly-L-Lysin-grafted-PolyEthylenGlycols (PLL-g-PEG) (PLL-g-PEG, PLL-g-PEG/PLL 1:1, PLL-g-PEG/PLL 10:1, PLL-g-PEG/PLL 1:10) and controls (glass, PLL, Mac-1 ligand GPIIb α). As expected, neutrophils adhered much less to the passivated surfaces (**figure S5a**) and were removed by simple washing. However, when not removed by washing they also performed NETosis on passivated surfaces (**supplementary figure S5c, movie M11 and M12**).

Supplementary figure S5: a, RISM images of fixed neutrophils on glass after one-time washing. Images show cells incubated with PMA for 30 min or cells left unstimulated. Cells on PLL-g-PEG (Poly-L-Lysine-grafted-PolyEthyleneGlycole) coated surfaces are barely adherent compared to neutrophils on glass or PLL coating. Scale = 10 μ m. **b**, Cell numbers after one-time washing of cells seeded for 30 min on different surfaces. In PLL-g-PEG coated wells only a few cells remain, compared to glass and PLL coating. These cells are still barely adherent as shown in a. $N=1$. **c**, PMA-induced NETosis (100 nM) performed on different surfaces. The amount of decondensed nuclei is independent of the surface/adhesive properties (surfaces of different passivation level, GPIIb/IIIa is the ligand of Mac-1 integrin). No washing step was included in this procedure. $N = 3$. Mean \pm SEM.

Furthermore, we also varied the substrate elasticity by using polyacrylamide gels of different stiffness and coated it with collagen I (see **figure R1**). Our data prove that our results hold true for different substrate elasticity, which could be the case in different *in vivo* compartments. We think that these results and others on LPS and fibrinogen coated surfaces (data not shown) are a separate topic by itself but wanted to show them because they further strengthen the conclusions of our paper as the reviewer pointed out.

[Figure R1 - Redacted]

Figure R1 (review only): Impact of substrate elasticity on NETosis. Our standard NETosis assay was performed on polyacrylamide (PAA) gels coated with collagen I (collaboration with Florian Rehfeldt/Göttingen University). The data show that PMA (5 nM) induced NETosis is not affected by substrate stiffness and the results on glass (also functionalized with collagen I) are similar. The stiffness covers the whole physiological range. $N=3$. Errors are SEM.

In summary, in our experimental system, stimulation by PMA outweighs the influence of adhesion. Indeed, our results indicate that NETosis can take place under non-adhesive conditions for example in the blood stream. However, there is evidence for a role of integrins, particularly Mac-1, in NETosis and our results certainly do not indicate that adhesion is not relevant. It seems very likely that adhesion especially for weaker stimuli may up- or downregulate the activation threshold of neutrophils in respect to NETosis and other cellular functions. After the cell “decides” to perform NETosis (end of phase 1) chromatin swelling drives the process and adhesion does not change that, which further supports the main conclusion of our paper.

References

1. Brinkmann, V., et al., *Automatic quantification of in vitro NET formation*. Front Immunol, 2012. **3**: p. 413.
2. van der Linden, M., et al., *Differential Signalling and Kinetics of Neutrophil Extracellular Trap Release Revealed by Quantitative Live Imaging*. Scientific Reports, 2017. **7**: p. 6529.
3. Gupta, S., et al., *A High-Throughput Real-Time Imaging Technique To Quantify NETosis and Distinguish Mechanisms of Cell Death in Human Neutrophils*. J Immunol, 2018. **200**(2): p. 869-879.
4. Fuchs, T.A., et al., *Novel cell death program leads to neutrophil extracellular traps*. J Cell Biol, 2007. **176**(2): p. 231-41.
5. van der Linden, M., et al., *Differential Signalling and Kinetics of Neutrophil Extracellular Trap Release Revealed by Quantitative Live Imaging*. Sci Rep, 2017. **7**(1): p. 6529.

6. Kettle, A.J., C.A. Gedye, and C.C. Winterbourn, *Mechanism of inactivation of myeloperoxidase by 4-aminobenzoic acid hydrazide*. *Biochemical Journal*, 1997. **321**(Pt 2): p. 503-508.
7. Forghani, R., et al., *Demyelinating diseases: myeloperoxidase as an imaging biomarker and therapeutic target*. *Radiology*, 2012. **263**(2): p. 451-60.
8. Kettle, A.J., et al., *Inhibition of myeloperoxidase by benzoic acid hydrazides*. *Biochem J*, 1995. **308** (Pt 2): p. 559-63.
9. Metzler, K.D., et al., *Myeloperoxidase is required for neutrophil extracellular trap formation: implications for innate immunity*. *Blood*, 2011. **117**(3): p. 953-9.
10. Metzler, K.D., et al., *A myeloperoxidase-containing complex regulates neutrophil elastase release and actin dynamics during NETosis*. *Cell Rep*, 2014. **8**(3): p. 883-96.
11. Wick, A.N., et al., *Localization of the primary metabolic block produced by 2-deoxyglucose*. *J Biol Chem*, 1957. **224**(2): p. 963-9.
12. Chen, W. and M. Gueron, *The inhibition of bovine heart hexokinase by 2-deoxy-D-glucose-6-phosphate: characterization by ³¹P NMR and metabolic implications*. *Biochimie*, 1992. **74**(9-10): p. 867-73.
13. Wick, A.N., D.R. Drury, and T.N. Morita, *2-Deoxyglucose; a metabolic block for glucose*. *Proc Soc Exp Biol Med*, 1955. **89**(4): p. 579-82.
14. Doiron, B., et al., *Transcriptional glucose signaling through the glucose response element is mediated by the pentose phosphate pathway*. *Journal of Biological Chemistry*, 1996. **271**(10): p. 5321-5324.
15. Colwell, D.R., J.A. Higgins, and G.S. Denyer, *Incorporation of 2-deoxy-D-glucose into glycogen. Implications for measurement of tissue-specific glucose uptake and utilisation*. *International Journal of Biochemistry & Cell Biology*, 1996. **28**(1): p. 115-121.
16. Ahadova, A., et al., *Dose-dependent effect of 2-deoxy-D-glucose on glycoprotein mannosylation in cancer cells*. *IUBMB Life*, 2015. **67**(3): p. 218-26.
17. Hansen, I.L., M.M. Levy, and D.S. Kerr, *The 2-deoxyglucose test as a supplement to fasting for detection of childhood hypoglycemia*. *Pediatr Res*, 1984. **18**(5): p. 490-5.
18. Olinger, A., P. Muley, and H. Tummala, *Effect of 2-Deoxyglucose on Colorectal Cancer Cell Lines*. *The Journal of Undergraduate Research*, 2013. **11**(1): p. 5.
19. Kipnis, D.M. and C.F. Cori, *Studies of tissue permeability. V. The penetration and phosphorylation of 2-deoxyglucose in the rat diaphragm*. *J Biol Chem*, 1959. **234**(1): p. 171-7.
20. Lundgaard, I., et al., *Direct neuronal glucose uptake heralds activity-dependent increases in cerebral metabolism*. *Nat Commun*, 2015. **6**: p. 6807.
21. Azevedo, E.P., et al., *A Metabolic Shift toward Pentose Phosphate Pathway Is Necessary for Amyloid Fibril- and Phorbol 12-Myristate 13-Acetate-induced Neutrophil Extracellular Trap (NET) Formation*. *J Biol Chem*, 2015. **290**(36): p. 22174-83.
22. Rodriguez-Espinosa, O., et al., *Metabolic requirements for neutrophil extracellular traps formation*. *Immunology*, 2015. **145**(2): p. 213-24.
23. Bowler, M.W., et al., *How azide inhibits ATP hydrolysis by the F-ATPases*. *Proc Natl Acad Sci U S A*, 2006. **103**(23): p. 8646-9.
24. Yoshikawa, S., et al., *Redox-coupled crystal structural changes in bovine heart cytochrome c oxidase*. *Science*, 1998. **280**(5370): p. 1723-9.
25. Lieber, C.S. and L.M. DeCarli, *Reduced nicotinamide-adenine dinucleotide phosphate oxidase: activity enhanced by ethanol consumption*. *Science*, 1970. **170**(3953): p. 78-80.
26. Rigo, A., P. Viglino, and G. Rotilio, *Polarographic determination of superoxide dismutase*. *Anal Biochem*, 1975. **68**(1): p. 1-8.

27. Pitanga, T.N., et al., *Neutrophil-derived microparticles induce myeloperoxidase-mediated damage of vascular endothelial cells*. BMC Cell Biol, 2014. **15**: p. 21.
28. Colin, F. and P. Quevauviller, *Monitoring of water quality: The contribution of advanced technologies*. 1998: Elsevier.
29. Bodine, J.H., *The Action of Sodium Azide Upon the Oxygen Uptake of Mitotically Active and Blocked Embryos*. Journal of Cellular and Comparative Physiology, 1950. **35**(3): p. 461-479.
30. VanUffelen, B.E., et al., *Sodium azide enhances neutrophil migration and exocytosis: involvement of nitric oxide, cyclic GMP and calcium*. Life Sci, 1998. **63**(8): p. 645-57.
31. Palmer, L.J., *NEUTROPHIL EXTRACELLULAR TRAPS IN PERIODONTITIS*, PhD thesis. University of Birmingham, 2010.
32. Lane, T.A. and G.E. Lamkin, *A reassessment of the energy requirements for neutrophil migration: adenosine triphosphate depletion enhances chemotaxis*. Blood, 1984. **64**(5): p. 986-93.
33. Bengtsson, T., K. Orselius, and J. Wettero, *Role of the actin cytoskeleton during respiratory burst in chemoattractant-stimulated neutrophils*. Cell Biol Int, 2006. **30**(2): p. 154-63.

Reviewers' Comments:

Reviewer #1:

Remarks to the Author:

The authors wrote an extensive "reply to the reviewers" and, in spite, of its length it does not reply to two main points. First, the use of inhibitors is difficult to interpret since, some of them like are unspecific like ABAH and others are affect pathways that are too central to the viability of the cell to make any interpretation possible, case in point is the use of 2-Deox-Gluc that affects glycolysis and could be inhibiting ROS production (through NADPH). Experiments to test this should be included.

Secondly, and more important, the data presented in the manuscript is consistent with other models where chromatin swelling is not a cause, but a consequence of the NETotic process. Case in point is the recent publication by Amulic et al (reference 55 in the manuscript) that postulates a mechanism aching to mitosis as the trigger of nuclear vesiculation. Indeed, if Amulic observation are correct (and as far as this reviewer is aware of, that paper included genetic data), the data in this manuscript would indicate that the swelling of chromatin is the consequence of lack of nuclear membrane restrain.

Reviewer #2:

Remarks to the Author:

The authors have very well revised their manuscript. They have introduced additional controls and rewritten the text to make the story very accurate and convincing. In my opinion they have commented well to all referees' concerns, and I am strongly supportive of publication as is.

Reviewer #3:

Remarks to the Author:

The authors have addressed all my concerns, going well beyond what was asked. The new data on weak adhesion and soft substrates is particularly interesting. I recommend publication as is.

Detailed response to reviewer 1 (review comment 2)

Reviewer comments are shown in blue/italic. Our responses in black.

“The authors wrote an extensive “reply to the reviewers” and, in spite, of its length it does not reply to two main points. First, the use of inhibitors is difficult to interpret since, some of them like are unspecific like ABAH”

ABAH is a specific inhibitor of MPO. This compound has been well characterized in the literature^{1,2}. In this very function ABAH has been used in some of the most important publications of the NETosis field³⁻⁶. The criticism of ABAH as a specific inhibitor of MPO therefore puts many of the fundamental conclusions of the last years regarding NET biology in question. As asked by the reviewer, we have again shown the effect of ABAH on MPO (supplementary figure S7). Of course, as with any chemical substance, one cannot completely exclude that this compound may have some unknown off-target effects.

However, we feel that it is way beyond the scope of this and any other paper to rule out that well-established methods/molecules have no possible (unknown) side effects whatsoever, especially when a multitude of complementary methods and controls are used to rule out off-target effects.

“and others are affect pathways that are too central to the viability of the cell to make any interpretation possible,

As previously asked by the reviewer, we have carefully assessed the influence of all compounds used in our cellular system on neutrophil viability (supplementary figure 7e). In our revisions, we clearly show that viability is not influenced, as claimed by the reviewer. It is true that some of the inhibitors such as sodium azide inhibit central functions of the cell, which was exactly the point we are trying to make. It is even more striking that such a potent inhibitor does not have any effect on NETosis in phase 2 any more (figure 3b), which exactly proves our hypothesis that the second phase of NETosis is driven by chromatin swelling.

“case in point is the use of 2-Deox-Gluc that affects glycolysis and could be inhibiting ROS production (through NADPH). Experiments to test this should be included.”

We feel that it is very unfortunate that reviewer 1 here raises novel questions about a biochemical side-effect which he had previously not voiced. Testing for ROS inhibition by 2-Deox-Gluc would have been a very simple assay, which could have been addressed alongside all the other enzymatic controls requested by reviewer 1 (supplementary figure S9). This assay could still be done by us within a matter of days.

Indeed, it has been already suggested that 2-Deox-gluc affects the pentose phosphate pathway and therefore also the generation of ROS through NADPH production in context of NETosis⁷.

However, proving or disproving an influence on ROS production would hardly affect our conclusions. 2-Deox-Gluc, like all the other inhibitors used in our manuscript, does not influence NETosis any more when added after the point of no return (start of phase 2 e.g. figure 3b).

Apparently, reviewer 1 is still reluctant to accept that it is not the specific function of the used inhibitors which is essential to our conclusion, but the fact that none of the inhibitors, whether they are specific or unspecific and whether they inhibit ROS production, energy metabolism or only specific enzymes important for NETosis, have an effect in phase 2 of NETosis.

However, to refute the doubts of reviewer 1 regarding the use of inhibitors in general, we previously already pointed out that we also use several absolutely non-invasive techniques to strengthen our conclusions, such as modification of the temperature (figure 3c) during the experiments and the measurement of ATP levels in the cells (figure 3a, supplementary figure S7b,c). The latter of which was added during the revision as a novel non-interventional method. Therefore, this criticism of the use of known biochemical inhibitors in general seems, with all due respect, far-fetched.

“Secondly, and more important, the data presented in the manuscript is consistent with other models where chromatin swelling is not a cause, but a consequence of the NETotic process.”

The complex process of NETosis cannot be reduced to such a generalization. The question we are asking is not whether chromatin expansion (alone) or biochemical pathways (alone) constitute NETosis and we believe that our manuscript presents a much more advanced view of the whole process and of the different phases of NETosis, in particular.

Indeed, there is absolutely no doubt that signalling processes and biochemical alterations are crucial to initiate NETosis: During the fundamentally important phase 1 chromatin, the nuclear envelope and many other components of the cell are enzymatically modified to enable chromatin expansion in phase 2.

We in no way claim that chromatin swelling is the only process important for NETosis, yet we show for the first time that from a certain point on, chromatin expansion becomes the main driving force in a very complex process, which is a truly novel finding. In that way, our data is, as the reviewer has again pointed out, consistent with current models of NETosis, but advances our understanding of this process significantly.

“Case in point is the recent publication by Amulic et al (reference 55 in the manuscript) that postulates a mechanism acting to mitosis as the trigger of nuclear vesiculation.

Indeed, if Amulic observation are correct (and as far as this reviewer is aware of, that paper included genetic data), the data in this manuscript would indicate that the swelling of chromatin is the consequence of lack of nuclear membrane restrain.”

We are well aware of the data shown by Amulic et al. and have, in fact, very recently been able to discuss our data and the good correlation with his own findings with him in person. We have no doubt that the data he shows is correct and very fascinating. However, we do not agree with the conclusions reviewer 1 draws from this publication and feel that he has cited this paper in a wrong way.

After carefully rereading the work of Amulic et al. we were not able to find the above-mentioned statement (“that the swelling of chromatin is the consequence of lack of nuclear membrane restrain”) in this paper. On the contrary, while Amulic et al. certainly shows a modification of the nuclear envelope early on (lamin phosphorylation as early as 30 min after PMA stimulation), he shows nuclear envelope breakdown approx. after 2 hours after PMA stimulation, right before NETs are released and where the chromatin is already expanded and the nucleus has lost its lobulated form. This is a time-point well after the time-point we have identified for the rupture of the nuclear envelope (at the end of phase 1, see figure 2c)!

In line with this, Amulic et al. also states that delobulation of the nucleus is a process that begins before, even as the cell prepares to release the NET. In light of our own data this would suggest that rupture of the nuclear envelope, as we see it, is a necessary event to release pressure that allows the expansion of the chromatin well before the final vesiculation and degradation of the nuclear envelope.

See also Amulic et al., *Developmental Cell*, 11/2017, Figure legend 1c):

“nonstimulated neutrophils have a lobulated nucleus, but this lobulation is lost as they prepare to release NETs. Approximately 2 hr after PMA stimulation, the nuclear envelope starts to disintegrate, allowing nuclear material to mix with the contents of the granules and the cytoplasm before being released into the extracellular space.”

The phosphorylation of nuclear lamins early on (at 30 min) described in the same paper, which putatively disrupts the structural rigidity of the nuclear envelope (Amulic et al., *Developmental Cell*, 11/2017), is also absolutely in line with our data, as this biochemical modification would allow for an easier rupture (and later for the vesiculation) of the nuclear envelope.

Thus, the conclusion of reviewer 1 that swelling of chromatin is the consequence of lack of nuclear membrane restrain cannot be derived from the paper mentioned by him.

We have added these aspects to our main document (see results, page 5):

“While, to our knowledge, this presents the first description of early rupture events of the nuclear envelope, previous publications have described the modification of nuclear lamins by phosphorylation as an early event, which would affect rigidity and could facilitate the here-described breakage of the nuclear envelope⁸. It should be noted that the breakage of the nuclear envelope appears to be a distinct process from the previously described dissolution of the nuclear envelope, which is a hallmark of late stages of NETosis^{13,8}.”

And to the discussion (page 16):

“We conclude that NETosis is a highly organized process with a first phase (P1) that is governed by biochemical modifications including histone citrullination and phosphorylation of lamins⁸ that prepare the cell for later mechanical changes.”

In conclusion, we feel that the two novel points raised by reviewer 1 are neither scientifically correct nor substantial for the results and the conclusions presented in our manuscript.

References

1. Kettle AJ, Gedye CA, Winterbourn CC. Mechanism of inactivation of myeloperoxidase by 4-aminobenzoic acid hydrazide. *Biochemical Journal* 1997; **321**(Pt 2): 503-8.
2. Kettle AJ, Gedye CA, Hampton MB, Winterbourn CC. Inhibition of myeloperoxidase by benzoic acid hydrazides. *Biochemical Journal* 1995; **308**(Pt 2): 559-63.
3. Metzler KD, Fuchs TA, Nauseef WM, et al. Myeloperoxidase is required for neutrophil extracellular trap formation: implications for innate immunity. *Blood* 2011; **117**(3): 953-9.
4. Metzler KD, Goosmann C, Lubojemska A, Zychlinsky A, Papayannopoulos V. A myeloperoxidase-containing complex regulates neutrophil elastase release and actin dynamics during NETosis. *Cell reports* 2014; **8**(3): 883-96.
5. Papayannopoulos V, Metzler KD, Hakkim A, Zychlinsky A. Neutrophil elastase and myeloperoxidase regulate the formation of neutrophil extracellular traps. *J Cell Biol* 2010; **191**(3): 677-91.
6. Kenny EF, Herzig A, Kruger R, et al. Diverse stimuli engage different neutrophil extracellular trap pathways. *eLife* 2017; **6**.
7. Azevedo EP, Rochael NC, Guimaraes-Costa AB, et al. A Metabolic Shift toward Pentose Phosphate Pathway Is Necessary for Amyloid Fibril- and Phorbol 12-Myristate 13-Acetate-induced Neutrophil Extracellular Trap (NET) Formation. *The Journal of biological chemistry* 2015; **290**(36): 22174-83.
8. Amulic B, Knackstedt SL, Abu Abed U, et al. Cell-Cycle Proteins Control Production of Neutrophil Extracellular Traps. *Developmental cell* 2017.